# Harnessing the Power of Neural Operators with Automatically Encoded Conservation Laws

## Abstract

Neural operators (NOs) have emerged as effective tools for modeling complex physical systems in scientific machine learning. In NOs, a central characteristic is to learn the governing physical laws directly from data. In contrast to other machine learning applications, partial knowledge is often known a priori about the physical system at hand whereby quantities such as mass, energy and momentum are exactly conserved. Currently, NOs have to learn these conservation laws from data and can only approximately satisfy them due to finite training data and random noise. In this work, we introduce conservation law-encoded neural operators (clawNOs), a suite of NOs that endow inference with automatic satisfaction of such conservation laws. ClawNOs are built with a divergence-free prediction of the solution field, with which the continuity equation is automatically guaranteed. As a consequence, clawNOs are compliant with the most fundamental and ubiquitous conservation laws essential for correct physical consistency. As demonstrations, we consider a wide variety of scientific applications ranging from constitutive modeling of material deformation, incompressible fluid dynamics, to atmospheric simulation. ClawNOs significantly outperform the state-of-the-art NOs in learning efficacy, especially in small-data regimes.

## 1 Introduction

Deep neural networks (DNN) have achieved tremendous progress in areas where the underlying laws are unknown. In computer vision applications convolutional neural networks have been very popular even though it is unclear how the lower-dimensional manifold of "valid" images is parameterized (He et al., 2016; Ren et al., 2015; Krizhevsky et al., 2012). That means, such a manifold has to be discovered in a purely data-driven way by feeding the network vast amounts of data. Another recent popular application of DNNs is the modeling and calibration process of physics-based problems from experimental measurements (Ranade et al., 2021; Schmidt & Lipson, 2009; Çolak, 2021; Jin et al., 2023; Vinuesa et al., 2023). A wide range of physical applications entail the learning of solution operators, i.e., the learning of infinite dimensional function mappings between any parametric dependence to the solution field. A prototypical instance is the case of modeling fluid dynamics, where the initial input needs to be mapped to a temporal sequence of flow states. Like the time integration in numerical modelling, an operator is required that takes the current flow state and maps it to the predicted state one time step later. To this end, the neural operator (NO) (Anandkumar et al., 2020; Li et al., 2020c; Lu et al., 2019; Gupta et al., 2021; Cao, 2021) is introduced, which learns a surrogate mapping between function spaces with resolution independence as well as generalizability to different input instances. These facts make NOs excellent candidates in discovering models for complex physical systems directly from data.

Herein, we consider data-driven model discovery of physics-based problems using NOs (Li et al., 2021; Goswami et al., 2022b). In contrast to computer vision applications, physics-based applications are often at least partially constrained by well-known fundamental laws. As a famous example, the motion of a particle in an external potential should conserve energy and momentum, while the exact form of the potential or the expression for the momentum equations is still unknown and needs to be inferred from observations. However, most of the current NOs have been focused on a pure data-driven paradigm, which neglects these intrinsic conservation of fundamental physical laws in data. As a result, their performances highly rely on the quantity and coverage of available data.

To improve learning efficacy and robustness in small-data regimes, we propose to encode a series of fundamental conservation laws into the architecture of NOs. Their inference is then constrained to a physically-consistent manifold. Here, we focus on the conservation of mass or volume which leads to a continuity equation for divergence-free flow. The development is based on **two key innovations**. Firstly, once the output function of a NO is divergence-free, the continuity equation is automatically satisfied, guaranteeing conservation of volume and mass. Based on the concept of differential forms, the conservation law is embedded through building divergence-free output functions. Secondly, to evaluate the differential forms, we propose an additional linear layer in NOs, whose weights are pre-specified based on high-order numerical differentiations on the given grids. Given an input function and its values on given grids, this layer evaluates the spatial derivatives as weighted linear combinations of neighboring or global points, mapping the function to its approximated derivatives.

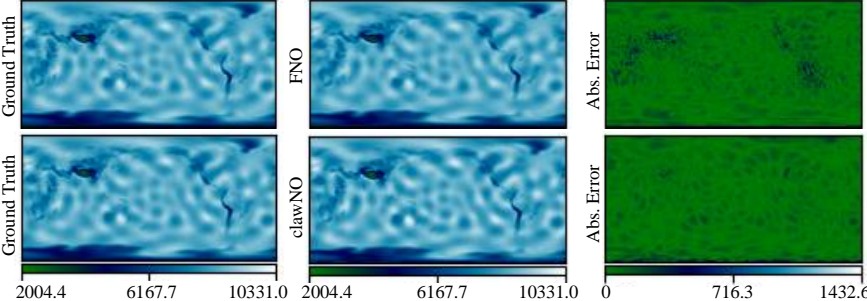

Figure 1: Predictability demo in atmospheric modeling. While FNO plausibly learns the wave propagation patterns, its relative L2 error is almost twice as big as that of clawNO (cf. Section 4.3), especially in the vicinity of mountains (Fig. 5). ClawNOs automatically satisfy the conservation law and improve physical consistency.

Compared to existing NO methods, the proposed architecture mainly carries three significant advantages. First, different from existing physics-informed neural operators (Goswami et al., 2022a;b; Li et al., 2021; Wang et al., 2021), our approach is readily applicable to learn physical systems directly from experimental measurements, since it only requires observed data pairs and does not rely on fully known governing equations. Second, the conservation laws in our approach are realized through built-in architectures, which constrain the output function to a physically consistent manifold independent of noisy measurement and/or scarce data coverage. Third, our architecture is designed to encode general conservation laws at large. Besides the conservation of mass, linear/angular momentum as demonstrated in our examples, it is readily applicable to the conservations of energy and electric charge, and can be easily scaled up to higher dimensions. In summary, the main contributions of our work are:

- We propose clawNO, a novel neural operator architecture to learn complex physical systems with conservation laws baked into the design.
- Our architecture is realized by adding an additional layer that employs numerical differentiation in evaluating the spatial and temporal derivatives as a weighted linear combination of neighboring points. As a result, our design is readily applicable as an add-on to any neural operator architecture, with comparable network size and computational cost.
- ClawNO only requires data pairs and does not rely on *a priori* domain knowledge, while the guaranteed conservation laws improve the learning efficacy, especially in small-data regime.

## 2 BACKGROUND AND RELATED WORK

**Learning hidden physics.** Learning and predicting complex physics directly from data is ubiquitous in many scientific and engineering applications (Ghaboussi et al., 1998; 1991; Carleo et al., 2019; Karniadakis et al., 2021; Zhang et al., 2018; Cai et al., 2022; Pfau et al., 2020; He et al., 2021; Besnard et al., 2006). Amongst these real-world physical problems, the underlying governing laws largely remain unknown, and machine learning models act to discover hidden physics from data through training. Successful examples include graph neural networks (GNNs) in discovering molecular properties (Wieder et al., 2020; Wu et al., 2023a), neural ODEs in constructing strain energy formulation for materials (Tac et al., 2022), and CNNs in steady flow approximation (Guo et al., 2016), etc. In this work, we focus on learning hidden physics from measurements with fundamental physical laws enforced. Examples include learning material deformation model from experimental measurements, where the mass should be conserved, but the constitutive law remains hidden.

**Neural operator learning.** Among others, NOs possess superiority in discovering physical laws as function mappings. Contrary to classical neural networks (NNs) that operate between finite-dimensional Euclidean spaces, NOs are designed to learn mappings between infinite-dimensional function spaces (Li et al., 2020a;b;c; You et al., 2022a; Ong et al., 2022; Cao, 2021; Lu et al., 2019; 2021; Goswami et al., 2022a; Gupta et al., 2021). As a result, NOs are often employed to manifest the mapping between spatial and/or spatio-temporal data pairs. Compared with classical NNs, the most notable advantages of NOs are their resolution independence and generalizability to different input instances. Moreover, NOs only require data with no knowledge on the underlying governing laws. All these advantages make NOs promising tools to learn hidden physics directly from data.

Despite the aforementioned advances, purely data-driven NOs still suffer from the data challenge: they require a large set of paired data, which is prohibitively expensive in many engineering applications. To resolve this challenge, physics-informed neural operator (PINO) (Li et al., 2021) and physics-informed DeepONets (Goswami et al., 2022a; Wang et al., 2021) are introduced, where a PDE-based loss is added to the training loss as a penalization term. However, these approaches often require *a priori* knowledge of all the underlying physics laws (in the form of governing PDEs). Moreover, the penalization term only imposes the these knowledge as a weak constraint.

**Imposing partial physical laws in NNs.** Several NN models have pursued the direction of imposing partial physics knowledge into the architecture of the model, with the goal of improving efficiency in learning hidden physics. A popular approach is to design equivariant NNs (Satorras et al., 2021; Müller, 2023; Liu et al., 2023; Helwig et al., 2023), which enforces symmetries in a Lagrangian represented by NNs and guarantees that the discovered physical laws do not depend on the coordinate system used to describe them. As pointed out by Sarlet & Cantrijn (1981); Noether (1971), such symmetries induce the conservation of linear and angular momentum in the learned physical system. Towards the conservation law of other quantities, the authors in Keller & Evans (2019); Sturm & Wexler (2022) consider the temporal concentration changes of multiple species. They find that conserving mass via a balancing operation improves the accuracy and physical consistency of NNs. In Richter-Powell et al. (2022), a divergence-free perspective has been considered to impose more general conservation laws in the framework of physics-informed neural networks (PINNs) (Raissi et al., 2019; Cai et al., 2021) with the aid of automatic differentiation to evaluate the derivatives directly. The same problem is also tackled by Négiar et al. (2022); Hansen et al. (2023), where the conservation law is enforced by projecting the solution to the constrained space (Négiar et al., 2022) or encoded by taking the predictive update following an integral form of the full governing equation (Hansen et al., 2023). However, in all these conservation law-informed neural networks the full governing PDEs are employed to formulate the loss function or the predictive updates/projections, and hence they are not applicable to hidden physics discovery.

## 3 CLAWNO: CONSERVATION LAW ENCODED NEURAL OPERATOR

We consider the learning of complex physical systems based on a number of observations of the input function values $\mathbf{f}_i(\mathbf{x}) \in \mathbb{F}(\Omega; \mathbb{R}^{p_f})$ and the corresponding output function values $\mathbf{u}_i(\mathbf{x}) \in \mathbb{U}(\Omega; \mathbb{R}^{p_u})$. Here, $i$ denotes the sample index, $\Omega \in \mathbb{R}^p$ is the bounded domain of interest, and $\mathbb{F}$ and $\mathbb{U}$ describe the Banach spaces of functions taking values in $\mathbb{R}^{p_f}$ and $\mathbb{R}^{p_u}$, respectively. To model the physical laws of such a system, we aim to learn the intrinsic operator $\mathcal{G}: \mathbb{F} \to \mathbb{U}$, that maps the input function $\mathbf{f}(\mathbf{x})$ to the output function $\mathbf{u}(\mathbf{x})$. In other words, given the measurements of $\mathbf{f}_i$ and $\mathbf{u}_i$ on a collection of points $\chi = \{\mathbf{x}_j\}_{j=1}^M \subset \Omega$, we seek to learn the physical response by constructing a parameterized surrogate operator of $\mathcal{G}$: $\tilde{\mathcal{G}}[\mathbf{f}; \theta](\mathbf{x}) \approx \mathbf{u}(\mathbf{x})$, where the trainable parameter set $\theta$ is obtained by solving the optimization problem:

$$\min_{\theta \in \Theta} \mathcal{L}(\theta) := \min_{\theta \in \Theta} \sum_{i=1}^N \frac{1}{N} [C(\tilde{\mathcal{G}}[\mathbf{f}_i; \theta], \mathbf{u}_i)] \approx \min_{\theta \in \Theta} \frac{1}{N} \sum_{i=1}^N \frac{\sum_{j=1}^M \left|\left| \tilde{\mathcal{G}}[\mathbf{f}_i; \theta](\mathbf{x}_j) - \mathbf{u}_i(\mathbf{x}_j) \right|\right|^2}{\sum_{j=1}^M ||\mathbf{u}_i(\mathbf{x}_j)||^2}. \qquad (1)$$

Here, $C$ denotes the cost functional that is often taken as the relative mean squared error.

Although our approach is agnostic to any NO architecture, for the purpose of demonstration we consider the generic integral neural operators (Li et al., 2020b;c; You et al., 2022a;b) as the base models. In this context, an $L$-layer NO has the following form:

$$\tilde{\mathcal{G}}[\mathbf{f}; \theta](\mathbf{x}) := \mathcal{Q} \circ \mathcal{J}_L \circ \cdots \circ \mathcal{J}_1 \circ \mathcal{P}[\mathbf{f}](\mathbf{x}), \qquad (2)$$

where $\mathcal{P}$, $\mathcal{Q}$ are shallow-layer NNs that map a low-dimensional vector into a high-dimensional vector and vice versa. Each intermediate layer, $\mathcal{J}_l$, consists of a local linear transformation operator $\mathcal{W}_l$, an integral (nonlocal) kernel operator $\mathcal{K}_l$, and an activation function $\sigma$. The architectures of

NOs mainly differ in the design of their intermediate layer update rules. As two popular examples, when considering the problem with structured domain $\Omega$ and uniform grid set $\chi$, a Fourier neural operator (FNO) is widely used, where the integral kernel operators $\mathcal{K}_l$ are linear transformations in frequency space. As such, the $(l+1)$th-layer feature function $\mathbf{h}(\mathbf{x}, l + 1)$ is calculated based on the $l$th-layer feature function $\mathbf{h}(\mathbf{x}, l)$ via:

$$\mathbf{h}(\mathbf{x}, l + 1) = \mathcal{J}_l^{FNO}[\mathbf{h}(\mathbf{x}, l)] := \sigma(W_l \mathbf{h}(\mathbf{x}, l) + \mathcal{F}^{-1}[A_l \cdot \mathcal{F}[\mathbf{h}(\cdot, l)]](\mathbf{x}) + \mathbf{c}_l), \tag{3}$$

where $W_l$, $\mathbf{c}_l$ and $A_l$ are matrices to be optimized, $\mathcal{F}$ and $\mathcal{F}^{-1}$ denote the Fourier transform and its inverse, respectively. On the other hand, the graph neural operator (GNO) is often employed on general irregular domains and grids, where the intermediate layer is invariant with respect to $l$, i.e., $\mathcal{J}_1 = \cdots, \mathcal{J}_L := \mathcal{J}^{GNO}$, with the update of each layer given by

$$\mathbf{h}(\mathbf{x}, l + 1) = \mathcal{J}^{GNO}[\mathbf{h}(\mathbf{x}, l)] := \sigma(W \mathbf{h}(\mathbf{x}, l) + \int_\Omega \kappa(\mathbf{x}, \mathbf{y}, \mathbf{f}(\mathbf{x}), \mathbf{f}(\mathbf{y}); \omega) \mathbf{h}(\mathbf{y}, l) d\mathbf{y} + \mathbf{c}). \tag{4}$$

Here, $\kappa(\mathbf{x}, \mathbf{y}, \mathbf{f}(\mathbf{x}), \mathbf{f}(\mathbf{y}); \omega)$ is a tensor kernel function that takes the form of a (usually shallow) NN with parameter $\omega$. In equation 4, the integral operator is often evaluated via a Riemann sum approximation $\int_\Omega \kappa(\mathbf{x}, \mathbf{y}, \mathbf{f}(\mathbf{x}), \mathbf{f}(\mathbf{y}); \omega) \mathbf{h}(\mathbf{y}, l) d\mathbf{y} \approx \frac{1}{M} \sum_{j=1}^M \kappa(\mathbf{x}, \mathbf{x}_j, \mathbf{f}(\mathbf{x}), \mathbf{f}(\mathbf{x}_j); \omega) \mathbf{h}(\mathbf{x}_j, l)$, realized through a message passing GNN architecture on a fully connected graph.

## 3.1 BUILT-IN DIVERGENCE-FREE PREDICTION

Intuitively, a conservation law states that a quantity of a physical system does not change as the system evolves over time. The most well-known examples of conserved quantities include mass, energy, linear momentum, angular momentum, etc. Mathematically, a conservation law can be expressed as a continuity equation defining the relation between the amount of the quantity and the "transport" of that quantity:

$$\frac{\partial \rho}{\partial t}(\mathbf{x}, t) + \nabla_\mathbf{x} \cdot \mu(\mathbf{x}, t) = 0, \quad \text{for } (\mathbf{x}, t) \in \Omega \times [0, T], \tag{5}$$

where $\rho : \mathbb{R}^{p+1} \to \mathbb{R}^+$ is the volume density of the quantity to be conserved, $\mu : \mathbb{R}^{p+1} \to \mathbb{R}^p$ is the flux describing how this quantity flows, and $\nabla_\mathbf{x}$ denotes the gradient operator with respect to the spatial dimensions. Equation 5 states that the amount of the conserved quantity within a volume can only change by the amount of the quantity that flows in and out of the volume. Taking the mass conservation law as an instance, $\rho$ is the fluid's density, and $\mathbf{u}$ is the velocity at each point, and then the mass flux becomes $\mu = \rho \mathbf{u}$. When considering the (quasi-)static problem, i.e., $\rho$ is independent of time, we have $\nabla_\mathbf{x} \cdot \mathbf{u} = 0$, where the divergence is taken only over spatial variables. Such a divergence-free equation is also referred to as the condition of incompressibility in fluid dynamics. Moreover, we note that equation 5 can always be expressed as a divergence-free equation when considering both spatial and temporal dimensions. As a matter of fact, by taking $\mathbf{X} = (\mathbf{x}, t)$ and $\mathbf{U} = (\mu, \rho)$, equation 5 can be re-written as $\nabla_\mathbf{X} \cdot \mathbf{U} = 0$, there the gradient operator $\nabla_\mathbf{X}$ is taken with respect to both $\mathbf{x}$ and $t$. Then, the conservation law becomes equivalent to the static case, if we consider the augmented variable $\mathbf{X}$ on the augmented domain $\tilde{\Omega} := \Omega \times [0, T]$. Hence, in the following derivation we focus on the static case, with the goal of designing an NO architecture with the continuity equation $\text{div}(\mathbf{u}) := \nabla_\mathbf{x}(\mathbf{u}) = 0$ automatically guaranteed for its output function $\mathbf{u}$.

To construct divergence-free NOs, we start from differential forms in $\mathbb{R}^p$, following the derivations in e.g., Barbarosie (2011); Kelliher (2021). More details are provided in Appendix A. Generally, an object that may be integrated over a $k-$dimensional manifold is called a $k-$form, which can be expressed as $\mu = \sum_{1 \le i_1, i_2, \cdots, i_k \le p} \mu_{(i_1, \cdots, i_k)} dx_{i_1} \wedge \cdots \wedge dx_{i_k} := \sum_I \mu_I dx_I$. For instance, any scalar function $g(\mathbf{x}) \in C^\infty$ is a $0-$form, and $v = \sum_{i=1}^p v_i(\mathbf{x}) dx_i$, $v_i(\mathbf{x}) \in C^\infty$, is a $1-$form. Denote $d$ as the exterior derivative, which is an operation acting on a $k-$form and produces a $(k + 1)-$form (e.g., $dg(\mathbf{x}) = \sum_{i=1}^p \frac{\partial g}{\partial x_i} dx_i$), and the Hodge operator as $\star$, which matches to each $k-$form an $(p - k)-$form via $\star dx_I := (-1)^{\sigma(I, I^c)} dx_{I^c}$. Here, $I^c := \{1, \cdots, n\} \backslash I$ is the complement set of $I$, and $\sigma(I, I^c)$ is the sign of the permutation $(I, I^c)$. A simple calculation yields that $\star \star \mu = (-1)^{k(p-k)} \mu$ and $dd\mu = 0$ for any $k-$form $\mu$.

To see the connection of these definitions to our goal, we note that our output function, $\mathbf{u} : \mathbb{R}^p \to \mathbb{R}^p$, can be equivalently expressed as a $1-$form $\mathbf{u} = \sum_{i=1}^p u_i(\mathbf{x}) dx_i$. Then, the divergence of $\mathbf{u}$ can be expressed as $d \star \mathbf{u}$, and the above fundamental properties of the exterior derivative yields $\text{div}(\star d\mu) = d \star \star d\mu = dd\mu = 0$ for any $(p - 2)-$form $\mu$. That means, any vector field $\mathbf{u}$ satisfying $\mathbf{u} = \star d\mu$ is divergence-free. Our goal is therefore to parameterize the output function of NOs in the form of $\star d\mu$, or equivalently, as $\star d(\sum_{1 \le i_1, \cdots, i_{p-2} \le p} \mu_{(i_1, \cdots, i_{p-2})} dx_{i_1} \wedge \cdots \wedge dx_{i_{p-2}})) = \sum_{i=1}^p \sum_{j=1}^p \frac{\partial \mu_{ij}}{\partial x_j} dx_i$.

Here, $\mu_{ij}(\mathbf{x})$ is a function defined on $\Omega$ with $\mu_{ij}(\mathbf{x}) = \mu_{(i,j)^c}$ for $i < j$ and $\mu_{ij}(\mathbf{x}) = -\mu_{(j,i)^c}$ otherwise, as real-valued alternating functions. That means, taking any $\mu := [\mu_{ij}(\mathbf{x})]$ satisfies $\mu_{ij} = -\mu_{ji}$ (a $p$ by $p$ skew-symmetric matrix-valued function), we have $\text{div}(\star d\mu) = 0$, where $\star d\mu = [\text{div}(\mu_1), \cdots, \text{div}(\mu_p)]^T$ and $\mu_i$ stands for the $i$−th row of $\mu$.

**Remark:** The idea of representing the conservation-law informed solution into the divergence of a skew-symmetric matrix-valued function is theoretically studied in Barbarosie (2011); Kelliher (2021), and recently employed in Richter-Powell et al. (2022) with the PINNs architecture. However, we point out that our scope of work is substantially different from Richter-Powell et al. (2022) where the governing equation is given. In PINN architecture the NN is constructed to approximate the solutions of a particular governing equation, i.e., mapping from $\mathbf{x}$ to the solution $\mathbf{u}(\mathbf{x})$. As such, the differential forms are approximated with automatic differentiation in NNs. In our work, we focus on the hidden physics learning problem where the governing equation remains *unknown* and employ the neural operator model to learn the governing law as a function-to-function mapping directly from data. Besides the fact that we do not rely on an equation-based loss function, in neural operators the differential forms can no longer be approximated via automatic differentiation, and therefore we propose a pre-calculated numerical differentiation layer as will be elaborated below.

## 3.2 CLAWNO ARCHITECTURE AND IMPLEMENTATION

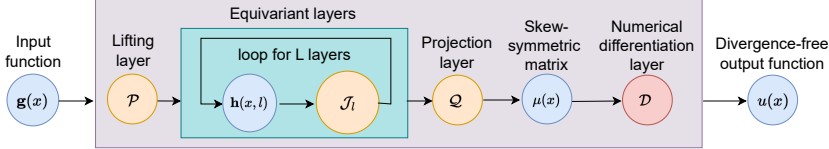

Figure 2: Illustration of the proposed clawNO architecture. We start from the input function $\mathbf{g}(x)$. After lifting, the high-dimensional latent representation goes through a series of iterative equivariant layers, then gets projected to a function space in the form of an antisymmetric matrix. Lastly, we employ numerical differentiation (layer $\mathcal{D}$ with pre-calculated weights) to obtain the target divergence-free output.

With the above analysis, we now construct an NO architecture with divergence-free output. Concretely, we propose to modify the architecture in equation 2 as:

$$\tilde{\mathcal{G}}[\mathbf{f}; \theta](\mathbf{x}) := \mathcal{D} \circ \mathcal{Q} \circ \mathcal{J}_L \circ \cdots \circ \mathcal{J}_1 \circ \mathcal{P}[\mathbf{f}](\mathbf{x}), \tag{6}$$

where the projection layer, $\mathcal{Q}$, maps the last layer feature vector $\mathbf{h}(\mathbf{x}, L)$ to a dimension $p(p-1)/2$ vector, such that each dimension in this vector stands for one degree of freedom in the skew symmetric matrix-valued function $\mu(\mathbf{x})$. Then, the last layer $\mathcal{D}$ takes $\mu$ as the input and aims to calculate its row-wise divergence $[\text{div}(\mu_1), \cdots, \text{div}(\mu_p)]^T$ as the output function $\mathbf{u}(\mathbf{x})$.

To formulate the last layer $\mathcal{D}$ there remains one more technical challenge: in NOs the output functions are provided on a set of discrete grid points, $\chi$, and therefore the derivatives are not readily evaluated. With the key observation that in classical numerical differentiation methods the derivatives are evaluated as weighted linear combinations of neighboring or global points, we propose to encode these weights in $\mathcal{D}$. As such, $\mathcal{D}$ will have pre-calculated weights and act as a numerical approximation of $[\text{div}(\mu_1), \cdots, \text{div}(\mu_p)]^T$. To demonstrate the idea, in the following we start with uniform grids $\chi$ and $\Omega$ being a periodic piece of $\mathbb{R}^p$, i.e., the torus $\mathbb{T}^p$, then discuss the scenarios with non-periodic domain and non-uniform grids.

Considering $\mathbb{T}^p := \prod_{i=1}^{p}[0, L^{(i)}]/\sim$ and $\chi := \left\{\left(\frac{k^{(1)}L^{(1)}}{N^{(1)}}, \cdots, \frac{k^{(p)}L^{(p)}}{N^{(p)}}\right) | k^{(i)} \in [0, N^{(i)}] \cap \mathbb{Z}\right\}$, each component of $\mu$ can then be approximated as a truncated Fourier series:

$$\mu_{jk}^{(N^{(1)}, \cdots, N^{(p)})}(\mathbf{x}) := \sum_{\xi_1 = -N^{(1)}/2}^{N^{(1)}/2 - 1} \cdots \sum_{\xi_p = -N^{(p)}/2}^{N^{(p)}/2 - 1} \hat{\mu}_{jk}(\xi_1, \cdots, \xi_p) e^{i2\pi\xi_1/L^{(1)}} \cdots e^{i2\pi\xi_p/L^{(p)}},$$

where $\hat{\mu}_{jk}(\xi_1, \cdots, \xi_p)$ is the Fourier coefficient of $\mu_{jk}$, calculated via $\mathcal{F}[\mu_{jk}]$. Taking derivatives of $\mu_{jk}^{(N^{(1)}, \cdots, N^{(p)})}$ and using them to approximate the derivative of $\mu_{jk}$, we have the following result:

**Theorem 3.1.** $\mathcal{D}$ is a $p(p-1)/2 \times M \times M \times p$ tensor with its parameters given by: $\mathcal{D}[\mu] = \star d\mu^{(N^{(1)}, \cdots, N^{(p)})} = [(\mathcal{D}[\mu])_1, \cdots, (\mathcal{D}[\mu])_p]^T$, where

$$(\mathcal{D}[\mu])_j = \sum_k \frac{\partial \mu_{jk}^{(N^{(1)}, \cdots, N^{(p)})}}{\partial x_k} = \sum_k \mathcal{F}^{-1}\left[i\frac{2\pi\xi_k}{L^{(k)}}\mathcal{F}[\mu_{jk}](\boldsymbol{\xi})\right], \tag{7}$$

*then the following error estimate holds true:*

$$||\star d\mu - \mathcal{D}[\mu]||_{L^\infty} = O(\Delta x^{m-1}),$$

*if $\mu$ has $m-1$ continuous derivatives for some $m \geq 2$ and a $m$-th derivative of bounded variation. In the special case when $\mu$ is smooth, a spectral convergence is obtained. Here, $M := \prod_{i=1}^{p} N^{(i)}$ denotes the total number of grids, and $\Delta x := \max_{i=1}^{p} L^{(i)}/N^{(i)}$ is the spatial grid size.*

*Proof.* Derivation of equation 7 is provided in Appendix D.1. The theorem is an immediate proposition of the Fourier spectral differentiation error estimate (see, e.g., Trefethen (2000, Page 34)). □

We now consider non-periodic domain and/or non-uniform grids. When the domain is non-periodic but the grids are uniform, we employ the Fourier continuation (FC) technique (Amlani & Bruno, 2016; Maust et al., 2022) which extends the non-periodic model output into a periodic function. In particular, the layer parameter for $\mathcal{D}$ in equation 7 is modified as: $\tilde{\mathcal{D}} := \mathcal{R} \circ \mathcal{D} \circ \mathbf{FC}$, where $\mathbf{FC}$ and $\mathcal{R}$ are the FC extension and restriction operators, respectively.

When the grids are non-uniform, the spectral method no longer applies and we seek to generate derivatives following the spirit of high-order meshfree methods (Bessa et al., 2014; Trask et al., 2019; Fan et al., 2023). In particular, considering the collection of measurement points $\chi = \{\mathbf{x}_i\}_{i=1}^{M}$, we seek to generate consistent quadrature rules of the form $\frac{\partial \phi}{\partial x_k}(\mathbf{x}_i) \approx \sum_{\mathbf{x}_j \in \chi \cap B_\delta(\mathbf{x}_i)} (\phi(\mathbf{x}_j) - \phi(\mathbf{x}_i))\omega_{i,j}^{(k)}$ for each $\mathbf{x}_i \in \chi$ and $k = 1, \cdots, p$. Here, $B_\delta(\mathbf{x}_i)$ denotes the neighborhood of radius $\delta$ of point $\mathbf{x}_i$, and $\omega_{i,j}^{(k)}$ is a collection of to-be-determined quadrature weights pre-calculated from the following optimization problem:

$$\min_{\omega_{i,j}^{(k)}} \sum_{\mathbf{x}_j \in \chi \cap B_\delta(\mathbf{x}_i)} |\omega_{i,j}^{(k)}|, \text{ s.t. } \frac{\partial \phi}{\partial x_k}(\mathbf{x}_i) = \sum_{\mathbf{x}_j \in \chi \cap B_\delta(\mathbf{x}_i)} (\phi(\mathbf{x}_j) - \phi(\mathbf{x}_i))\omega_{i,j}^{(k)}, \quad \forall \phi \in \mathbb{P}_m(\mathbb{R}^p), \quad (8)$$

where $\mathbb{P}_m(\mathbb{R}^p)$ is the space of $m$-th order polynomials ($m \geq 1$). Assume that $\chi$ is quasi-uniform, the size of $\chi \cap B_\delta(\mathbf{x})$ is bounded, the domain $\Omega$ satisfies a cone condition, and $\delta$ is sufficiently large, the above optimization problem has a solution (Wendland, 2004), and we further have:

**Theorem 3.2.** *Consider a fixed ratio of $\delta/\Delta x$ where $\Delta x := \sup_{\mathbf{x}_i \in \chi} \min_{\mathbf{x}_j \in \chi \backslash \mathbf{x}_i} ||\mathbf{x}_i - \mathbf{x}_j||_{L^2}$, the layer parameters of $\mathcal{D}$ can be formulated as: $\mathcal{D}[\mu] = [(\mathcal{D}[\mu])_1, \cdots, (\mathcal{D}[\mu])_p]^T$, where*

$$(\mathcal{D}[\mu])_j(\mathbf{x}_i) = \sum_k \sum_{\mathbf{x}_l \in \chi \cap B_\delta(\mathbf{x}_i)} (\mu_{jk}(\mathbf{x}_l) - \mu_{jk}(\mathbf{x}_i))\omega_{i,l}^{(k)}. \quad (9)$$

*The following error estimate holds true:*

$$||\star d\mu - \mathcal{D}[\mu]||_{L^\infty} = O(\Delta x^{m+1}),$$

*if $\mu$ has $m+1$ continuous derivatives.*

*Proof.* The proof follows a similar argument as in Levin (1998, Theorem 5), see Appendix D.2. □

The proposed architecture can be readily combined with the FNOs in equation 3 and GNOs in equation 4. Generally, when measurements are provided on regular domain and uniform grids, we consider FNOs together with the differentiation layer $\mathcal{D}$ approximated in equation 7 (or its FC-based extension). When the measurements are on non-uniform grids, GNOs can be employed with the differentiation layer $\mathcal{D}$ approximated via meshfree methods (see equation 9). Besides vanilla FNOs and GNOs, to further encode laws of physics we also consider their group equivariant versions as the base models. Namely, the invariant neural operator (INO) ((Liu et al., 2023)) for non-uniform grids and the group-equivariant FNO (G-FNO) ((Helwig et al., 2023)) for uniform grids. As such, the learnt laws of physics do not depend on the coordinate system used to describe them, together with the encoded conservation laws guaranteed.

## 4 EXPERIMENTS

We showcase the prediction accuracy and expressivity of clawNOs across a wide range of scientific problems, including elasticity, shallow water equations and incompressible Navier-Stokes equations. We compare the performance of clawNOs against a number of relevant machine learning techniques. In particular, we select INO (Liu et al., 2023) and GNO (Li et al., 2020a) as baselines in graph-based settings, whereas we choose FNO (Li et al., 2020c), G-FNO (Helwig et al., 2023), UNet (Ronneberger et al., 2015), LSM (Wu et al., 2023b), KNO (Xiong et al., 2023a;b), and UNO (Rahman et al., 2022) as baselines in datasets with uniform discretizations. We also compare with an additional non-NO baseline, Lagrangian neural networks (LNN) (Cranmer et al., 2020; Müller, 2023),

in the first experiment. The relative L2 error is reported as comparison metrics. We perform three replicates of all the experiments with randomly selected seeds, and the mean and standard deviation of the errors are reported. To guarantee a fair comparison, we make sure that the total numbers of trainable parameters in all models are on the same level, and report these numbers for each case. Additional details on the data generation and training strategies are provided in Appendix B.

## 4.1 INCOMPRESSIBLE NAVIER-STOKES EQUATION

We start from the incompressible Navier–Stokes equations in vorticity form, which is widely applied to the simulation of sub-sonic flows, especially in hydromechanical and turbulent dynamics modeling. The (known) mass conservation law and (hidden) governing equation are given by:

$$\nabla \cdot \mathbf{u} = 0 \,, \quad \partial_t w + \mathbf{u} \cdot \nabla w = \nu \Delta w + \mathbf{f} \,, \tag{10}$$

where $w$ and $\nu$ denote the vorticity and fluid viscosity, respectively, $\mathbf{f} = 0.1(sin(2\pi(x+y)) + cos(2\pi(x+y)))$ represents the external forcing term, and $\mathbf{u}$ is the velocity field that we aim to learn. The system is modeled on a square domain of $[0,1]^2$ and initialized with a random vorticity field $w_0$ sampled from a random Gaussian distribution. We consider two case studies. The first study predicts a rollout of $T = 20$ subsequent timesteps conditioned on the first $T_{in} = 10$ timesteps, which is consistent with the setting in G-FNOs (Helwig et al., 2023). In the second study, we predict a rollout of $T = 25$ timesteps conditioned on the first $T_{in} = 5$ timesteps. Compared with the first setting, this study aims to provide a longer-term extrapolation with less information from inputs. It is designed to further challenge the proposed model and baselines. Both studies are carried out in three datasets representing small, medium, and large data regimes.

Table 1: Test errors and the number of trainable parameters for the incompressible Navier–Stokes problem, where bold numbers highlight the best method. Rollouts are of length $T$ conditioned on the first $T_{in}$ time steps.

| Case | 2D models | #param | # of training samples | | |
|---|---|---|---|---|---|
| | | | 10 | 100 | 1000 |
| $T_{in} = 10, T = 20$ | clawGFNO | 853,231 | **12.10%±1.08%** | **4.76%±0.24%** | **1.21%±0.09%** |
| | clawFNO | 928,861 | 12.68%±0.85% | 5.17%±0.27% | 1.46%±0.11% |
| | GFNO-p4 | 853,272 | 29.56%±3.32% | 7.59%±0.34% | 3.11%±0.93% |
| | FNO | 928,942 | 22.57%±1.87% | 5.68%±0.41% | 1.47%±0.18% |
| | UNet | 920,845 | 56.81%±9.91% | 16.62%±2.47% | 6.07%±0.36% |
| | LSM | 1,211,234 | 31.79%±2.11% | 13.30%±0.15% | 5.52%±0.08% |
| | UNO | 1,074,522 | 21.48%±1.49% | 11.41%±0.38% | 5.29%±0.09% |
| | KNO | 890,055 | 20.26%±1.50% | 11.40%±3.12% | 9.09%±1.70% |
| | LNN | 1,056,768 | 25.73%±0.96% | 24.27%±3.16% | 21.36%±2.46% |
| $T_{in} = 5, T = 25$ | clawGFNO | 853,131 | **16.02%±0.57%** | **4.84%±0.49%** | **1.57%±0.18%** |
| | GFNO-p4 | 853,172 | 36.35%±4.47% | 10.67%±3.00% | 2.68%±0.60% |
| | FNO | 928,742 | 26.65%±5.31% | 6.61%±0.83% | 1.73%±0.23% |
| | UNet | 919,855 | 76.12%±14.04% | 34.45%±10.59% | 7.39%±0.83% |

**Ablation study** We first perform an ablation study in case 1 (i.e., $T = 20$ and $T_{in} = 10$) by adding the proposed divergence-free architecture to FNO and comparing its performance with the vanilla FNO and clawGFNO. The corresponding results are listed in Table 1, where we observe a consistent performance improvement with the proposed divergence-free architecture in all data regimes. In particular, by comparing clawGFNO to GFNO, we observe a boost in accuracy by 59.1%, 37.2% and 61.2% in small, medium and large data regimes, respectively. Our findings are consistent when we compare clawFNO to the vanilla FNO, where an enhancement by 43.8%, 9.0% and 0.7% is obtained, respectively. Therefore, in small training data regimes, imposing the conservation law has enhanced the data efficiency. This argument is further verified when comparing the distribution of test errors for clawGFNO and GFNO as demonstrated in Figure 3, from which one can see that clawGFNO has smaller errors on out-of-distribution test samples. Note that it is expected that the performance improvement of clawNOs becomes less pronounced as we march from the small data regime to the large data regime, as the baseline NOs are anticipated to learn the conservation laws from data as more data becomes available. The performance of clawNOs and the baseline NOs will eventually converge in the limit of infinite data. However, this is typically prohibitive as data generation is expensive in general, especially in the scientific computing community where one simulation can take hours to days to complete. On the other hand, clawGFNO further outperforms clawFNO in all data regimes, benefiting from the equivariant architecture of the model.

**Comparison against additional baselines** We choose to work with clawGFNO in the 2nd case study owing to its outstanding performance compared to clawFNO. Herein, we compare the performance of clawGFNO against additional baselines in Table 1. Among all baselines in case 1, KNO

has achieved the best performance in small data regime, while clawFNO and clawGFNO still outperform it by 37.4% and 40.3%, respectively. By investigating in the direction from small data regime to large data regime, clawGFNO achieves the best performance in all cases and beats the best baselines by 40.3%, 16.2% and 17.6% in case 1 and 39.9%, 26.8% and 9.1% in case 2. ClawGFNO also manifests its robustness as the learning task becomes more challenging across the two case studies, as compared to other baselines whose performance deteriorates notably. Additionally, in order to maintain a similar total number of trainable parameters, we reduce the latent width of GFNO from 20 to 11, leading to possible expressivity degradation in GFNO. In fact, when increasing the latent dimension of GFNO to 20 to match the dimension of FNO, on the large data regime of case 1 the test error is decreased to $1.19\% \pm 0.16\%$, achieving a comparable performance to FNO. Note also that LNN in case 1 seemingly cannot learn the correct solution as the test error remains above 20% across all three data regimes. This is due to the fact that LNN predicts the acceleration from current position and velocity, and then updates the velocity. Since this dataset is relatively sparse in time, it leads to large errors from temporal integration in LNNs.

## 4.2 RADIAL DAM BREAK

In this example, we explore the shallow water equations that are derived from the compressible Navier–Stokes equations, which find broad applications in tsunami and general flooding simulation. Specifically, we simulate a circular dam removal process where the water is initially confined in the circular dam and suddenly released due to the removal of the dam. The (known) mass conservation law and (hidden) governing PDE describing the system are as follows:

$$\partial_t h + \nabla \cdot (h\mathbf{u}) = 0 \,, \quad \partial_t (h\mathbf{u}) + \nabla \cdot \left( h\mathbf{u}\mathbf{u}^T \right) + \frac{1}{2} g\nabla h^2 + gh\nabla b = 0 \,, \tag{11}$$

where $g$ is the gravitational acceleration, $h$ describes the water height, $\mathbf{u}$ denotes the velocity field, $(h, h\mathbf{u})$ is the 3D divergence-free field we aim to learn, and $b$ denotes a spatially varying bathymetry. The simulation is performed on a square domain of $[-2.5, 2.5]^2$, and the initial height $h_0$ is specified as $h_0 = 2.0$ when $r < \sqrt{x^2 + y^2}$ and $h_0 = 1.0$ otherwise, with $r$ being the radial distance to the dam center, which is uniformly sampled from $(0.3, 0.7)$.

Table 2: Test errors and the number of trainable parameters for the radial dam break problem, where bold numbers highlight the best method. Rollouts are of length 24 conditioned on the first time step, following the setting of Helwig et al. (2023).

| 3D models | #param | # of training samples | | |
|---|---|---|---|---|
| | | 2 | 10 | 100 |
| clawFNO | 4,922,303 | **21.24%±1.17%** | **2.69%±0.04%** | 1.84%±0.01% |
| clawGFNO | 4,799,732 | 23.29%±3.10% | 2.87%±0.03% | **1.83%±0.01%** |
| FNO | 4,922,303 | 29.88%±0.17% | 4.04%±0.10% | 1.88%±0.05% |
| GFNO-p4 | 4,799,732 | 27.31%±2.80% | 3.80%±0.28% | 1.90%±0.07% |
| UNet | 5,080,113 | 338.08%±553.62% | 5.62%±0.19% | 2.98%±0.20% |

We report in Table 2 our experimental observations using clawGFNO and clawFNO, along with the comparison against FNO, GFNO, and UNet. ClawNOs continue to achieve the lowest test errors across all three selected datasets, where clawFNO shows the best performance in the small and medium data regimes and clawGFNO becomes slightly more superior in the large data regime. As the radial dam break problem is fairly simple and the solution exhibits strong symmetries, the performances of non-claw baselines quickly converge to clawNOs with only 100 training samples. Nevertheless, clawNOs consistently outstand the best non-claw baseline in performance by 22.2% in small data regime and 29.2% in medium data regime.

## 4.3 ATMOSPHERIC MODELING

In this example, we use the general circulation model SpeedyWeather.jl to simulate gravity waves in the Earth's atmosphere. The shallow water equations for relative vorticity $\zeta = \nabla \times \mathbf{u}$, divergence $\mathcal{D} = \nabla \cdot \mathbf{u}$, and the displacement $\eta$ from the atmosphere's rest height $H = 8500$ m are described as the given mass conservation law $\frac{\partial \eta}{\partial t} + \nabla \cdot (\mathbf{u}h) = 0$ together with other (hidden) governing laws:

$$\frac{\partial \zeta}{\partial t} + \nabla \cdot (\mathbf{u}(\zeta + f)) = 0 \,, \quad \frac{\partial \mathcal{D}}{\partial t} - \nabla \times (\mathbf{u}(\zeta + f)) = -\nabla^2 (\tfrac{1}{2}|\mathbf{u}|^2 + g\eta) \,. \tag{12}$$

The equations are solved in spherical coordinates with latitude $\theta \in [-\pi, \pi]$, longitude $\lambda \in [0, 2\pi]$ on a sphere of radius $R = 6371$ km. The layer thickness is $h = \eta + H - H_o$ with the Earth's orography $H_o = H_o(\lambda, \phi)$ and gravity $g = 9.81$ ms$^{-2}$. The Coriolis force uses the parameter $f = 2\Omega \sin(\theta)$ with $\Omega = 7.29 \cdot 10^{-5}$ s$^{-1}$ the angular frequency of Earth's rotation. The simulations start from rest, $\mathbf{u} = 0$, but with random waves in $\eta$ with wave lengths of about 2000 to 4000 km and maximum

amplitudes of 2000 m. These random initial waves propagate at a phase speed of $c_{ph} = \sqrt{gh}$ (about 300 ms$^{-1}$) around the globe, interacting non-linearly with each other and the underlying Earth's orography. For details on the model setup and the simulation see Appendix B.3.

Table 3: Test errors for the weather modeling problem and the number of trainable parameters (in millions), where bold numbers highlight the best method. Rollouts are of length 10 conditioned on the first time step.

| Model | clawFNO | FNO | GFNO | UNet |
|---|---|---|---|---|
| #param (M) | 49.57 | 49.57 | 53.70 | 51.58 |
| Test error | **6.64%±0.23%** | 12.54%±0.26% | 14.34%±0.55% | 20.89%±0.87% |

We list in Table 3 our experimental results along with comparisons against selected baselines. For this problem, since the spherical coordinate system in latitudes and longitudes stay the same and thus the equivariance property plays little role, we directly employ clawFNO as our representative clawNO model, as the latent dimension of clawFNO is much higher than that of clawGFNO for similar total numbers of model parameters. In other words, clawFNO possesses more expressivity power. This is rather important in atmospheric modeling at the entire globe scale, as the wave propagation patterns are more localized compared to other examples. In this context, clawFNO outperforms the best non-claw baseline by 47% in accuracy. Additionally, it is worth mentioning that the majority of the errors accumulate in the vicinity of mountains, as evidenced in the rollout demonstration in Figures 11-13 (compare to Fig. 5a). This is expected as the interaction with the orography causes wave reflection and dispersion through changes in the wave speed, which generally limits predictability more than in other regions. Furthermore, the training strategy (Appendix B.5) disregards wave numbers higher than 22 which, however, are more pronounced around rough orography. Our simulations resolve orography up to wave number 63 (Appendix B.3).

### 4.4 CONSTITUTIVE MODELING OF MATERIAL DEFORMATION

In this example we test the efficacy of the proposed clawNO in graph-based settings. Specifically, we consider the elastic deformation in materials, where the (hidden) constitutive law reads:

$$-\nabla \cdot \sigma = \mathbf{f}, \quad \sigma = \lambda(\nabla \cdot \mathbf{u})\mathbf{I} + \mu(\nabla\mathbf{u} + \nabla\mathbf{u}^T), \quad \mathbf{u}|_{\partial\Omega} = \mathbf{u}_D, \tag{13}$$

with Dirichlet boundary conditions, and the (known) mass conservation law $\nabla \cdot \mathbf{u} = 0$ imposed. Here $\sigma$ is the stress tensor, $\mathbf{u}$ denotes the displacement, $\mathbf{f}$ denotes the body load, $\lambda$ and $\mu$ are the Lame constants. Under this setting, the Dirichlet boundary condition and the body load are treated as inputs. To investigate the efficacy on irregular grids, we consider a circular domain of radius 0.4.

Table 4: Test errors and the number of trainable parameters for the material deformation problem, where bold numbers highlight the best method.

| 2D models | #param | # of training samples | | |
|---|---|---|---|---|
| | | 5 | 20 | 100 |
| clawINO | 4,747,521 | **8.54%±0.12%** | **5.88%±0.04%** | **3.92%±0.16%** |
| INO | 4,747,521 | 10.51%±0.61% | 6.66%±0.25% | 4.11%±0.64% |
| GNO | 18,930,498 | 10.02%±0.26% | 7.68%±0.05% | 5.82%±0.46% |

We report in Table 4 the numerical results of clawINO, along with the comparison against GNO and INO in various data regimes. Consistent with the findings on regular grids, clawINO outperforms other graph-based models in all data regimes, beating the best non-claw baseline by 14.8%, 11.7%, and 4.6% in the considered three data regimes, respectively. Note that the total number of parameters in GNO is roughly four times the number of parameters in INO-based models, as the parameters in INO-based models are layer-independent, which also results in an advantage in memory cost.

## 5 CONCLUSION

We introduce a series of clawNOs that explicitly bake fundamental conservation laws into the neural operator architecture. In particular, we build divergence-free output functions in light of the concept of differential forms and design an additional layer to recover the divergence-free target functions based on numerical differentiation. We perform extensive experiments covering a wide variety of challenging scientific machine learning problems, and show that it is essential to encode conservation laws for correct physical realizability, especially in small-data regimes. For the future work we will explore more fundamental laws in other application domains. We also point out that the idea of incorporating numerical PDE techniques into NOs can also be extended. For instance, the quadrature weights can provide a more accurate numerical integration in evaluating the cost functional, as compared to the current mean squared error widely employed in NOs.

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

## A    RELATED CONCEPTS OF DIFFERENTIAL FORMS

Differential forms aim to provide a unified approach to define integrands. In this section, we briefly review a few related mathematical concepts in the paper, which can be found in Richter-Powell et al. (2022); Do Carmo (1998); Weintraub (2014).

**Definition A.1.** *Let $k$ be a non-negative integer and $\mu(\mathbf{x}) = f(x_1, \cdots, x_p)$ be a smooth function on $\mathbb{R}^p$, a monomial $k-$form on $\mathbb{R}^p$ is an expression $\mu\, dx_{i_1} \wedge \cdots \wedge dx_{i_k} = \mu dx_I$, where $I := \{i_1, \cdots, i_k\}$. A $k-$form is a sum of monomial $k-$forms, which will be denoted as*

$$\mu = \sum_{1 \le i_1, i_2, \cdots, i_k \le p} \mu_{(i_1, \cdots, i_k)} dx_{i_1} \wedge \cdots \wedge dx_{i_k} := \sum_I \mu_I dx_I.$$

**Definition A.2.** *The wedge product, $\wedge$, is an associative operation on differential forms. When $\omega$ is a $k-$form and $\eta$ is a $l-$form, $\omega \wedge \eta$ is a $(k+l)-$form. Moreover, $\wedge$ satisfies the following properties:*

- *(Distribution Law) $(\omega_1 + \omega_2) \wedge \eta = \omega_1 \wedge \eta + \omega_1 \wedge \eta$, $\omega \wedge (\eta_1 + \eta_2) = \omega \wedge \eta_1 + \omega \wedge \eta_2$.*

- *(Associative Law) $(f\omega) \wedge \eta = f(\omega \wedge \eta)$.*

- *(Skew Symmetry) $\eta \wedge \omega = -\omega \wedge \eta$.*

*Here $\omega$, $\omega_1$, $\omega_2$ are $k-$forms, $\eta$, $\eta_1$, $\eta_2$ are $l-$forms, and $f$ is a function.*

**Definition A.3.** *The exterior derivative (or differential) operator $d$ maps a $k-$form to a $(k+1)-$form. In particular, given a $k-$form $\mu = \sum_{1 \le i_1 < i_2 < \cdots < i_k \le p} \mu_{(i_1, \cdots, i_k)} dx_{i_1} \wedge \cdots \wedge dx_{i_k}$, one has*

$$d\mu = \sum_{1 \le i_1, i_2, \cdots, i_k \le p} d\mu_{(i_1, \cdots, i_k)} dx_{i_1} \wedge \cdots \wedge dx_{i_k}$$

$$= \sum_{1 \le i_1, i_2, \cdots, i_k \le p} \sum_{1 \le l \le p} \frac{\partial \mu_{(i_1, \cdots, i_k)}}{\partial dx_l} dx_l \wedge dx_{i_1} \wedge \cdots \wedge dx_{i_k}.$$

*Moreover, the following properties hold for the exterior derivative operator:*

- *For each function $f$, $ddf = 0$, and $df \wedge df = 0$.*

- *For each $k-$form $\omega$, $dd\omega = 0$.*

- *For each function $f$ and $k-$form $\omega$, $d(f\omega) = df \wedge \omega + f d\omega$.*

**Definition A.4.** *The Hodge $\star$-operator on $\mathbb{R}^p$ is a function that takes a $k-$form to a $(p-k)-$form defined as follows:*

- *Let $I = \{i_1, \cdots, i_k\}$ be an ordered subset of $\{1, \cdots, p\}$ and $I^c = \{j_1, \cdots, j_{p-k}\}$ be its complement, ordered so that $dx_{i_1} \wedge \cdots dx_{i_k} \wedge dx_{j_1} \wedge \cdots dx_{j_{p-k}} = dx_1 \wedge \cdots \wedge dx_p$, then*

$$\star dx_I = \star(dx_{i_1} \wedge \cdots dx_{i_k}) := dx_{j_1} \wedge \cdots dx_{j_{p-k}} = dx_{I^c}.$$

- *Given any $k-$form $\mu = \sum_{1 \le i_1, i_2, \cdots, i_k \le p} \mu_{(i_1, \cdots, i_k)} dx_{i_1} \wedge \cdots \wedge dx_{i_k}$, then*

$$\star\mu := \sum_{1 \le i_1, i_2, \cdots, i_k \le p} \mu_{(i_1, \cdots, i_k)} \star (dx_{i_1} \wedge \cdots \wedge dx_{i_k}).$$

*Then, the following property holds for any $k-$form $\mu$:*

$$\star \star \mu = (-1)^{k(p-k)} \mu.$$

Given a vector-valued function on $\mathbb{R}^p$ $\mathbf{u}(\mathbf{x}) = [u_1(\mathbf{x}), \cdots, u_p(\mathbf{x})]$, its divergence $\text{div}(\mathbf{u}) = \sum_{i=1}^{p} \frac{\partial u_i}{\partial x_i}$ can be defined via exterior derivative:

$$d \star \sum_{i=1}^{p} u_i dx_i = d \sum_{i=1}^{p} u_i \star dx_i = d \sum_{i=1}^{p} (-1)^{i-1} u_i dx_{\{1,\cdots,n\}\setminus i}$$

$$= \sum_{i=1}^{p} (-1)^{i-1} du_i \wedge dx_{\{1,\cdots,n\}\setminus i} = \sum_{i=1}^{p} \frac{\partial u_i}{\partial x_i} dx_{\{1,\cdots,n\}}.$$

Therefore, one can denote the divergence of $\mathbf{u}$ as $d \star \mathbf{u}$. On the other hand, combining the fundamental properties of $\star$ and $d$ yields $\text{div}(\star d\mu) = d \star \star d\mu = (-1)^{k(p-k)} dd\mu = 0$ for any $k-$form $\mu$. Therefore, when $\mathbf{u} = \star d\mu$, where $\mu$ is a $(p-2)-$form, it is guaranteed to be divergence free. Since any $(p-2)-$form $\mu$ can be represented as

$$\mu = \sum_{1 \le i_1, \cdots, i_{p-2} \le p} \mu_{(i_1, \cdots, i_{p-2})} dx_{i_1} \wedge \cdots \wedge dx_{i_{p-2}} = \sum_{1 \le i,j \le p} \mu_{(i,j)} \star (dx_i \wedge dx_j),$$

where $\mu_{(i,j)} := (-1)^{\sigma(\{i,j\},\{i_1,\cdots,i_{p-2}\})} \mu_{(i_1,\cdots,i_{p-2})}$ when $\{i,j\} \cup \{i_1,\cdots,i_{p-2}\} = \{1,\cdots,p\}$. And note that the skew symmetry property of $\wedge$ yields $\mu_{(i,j)} := \mu_{ij} = -\mu_{ji}$. We further expand the representation of $\star d\mu$ as:

$$\star d\mu = \star \sum_{1 \le i,j \le p} d\mu_{(i,j)} \star (dx_i \wedge dx_j)$$

$$= \star \left( \sum_{1 \le i,j \le p} \frac{\partial \mu_{ij}}{\partial x_i} dx_i \wedge \star(dx_i \wedge dx_j) - \frac{\partial \mu_{ji}}{\partial x_j} dx_j \wedge \star(dx_i \wedge dx_j) \right)$$

$$= \star \left( \sum_{1 \le i,j \le p} \frac{\partial \mu_{ij}}{\partial x_i} dx_i \wedge \star(dx_i \wedge dx_j) + \frac{\partial \mu_{ij}}{\partial x_i} dx_i \wedge \star(dx_i \wedge dx_j) \right)$$

$$= 2 \star \sum_{1 \le i,j \le p} \frac{\partial \mu_{ij}}{\partial x_i} dx_i \wedge \star(dx_i \wedge dx_j) = 2 \star \sum_{1 \le i,j \le p} \frac{\partial \mu_{ij}}{\partial x_i} \star dx_j$$

$$= 2 \sum_{1 \le i,j \le p} \frac{\partial \mu_{ij}}{\partial x_i} \star \star dx_j = (-1)^{p-1} 2 \sum_{j} \left( \sum_{i} \frac{\partial \mu_{ij}}{\partial x_i} \right) dx_j.$$

That means, taking any $\mu := [\mu_{ij}(\mathbf{x})]$ satisfies $\mu_{ij} = -\mu_{ji}$ (a $p$ by $p$ skew-symmetric matrix-valued function), we have $\text{div}(\star d\mu) = 0$, where $\star d\mu = [\text{div}(\mu_1), \cdots, \text{div}(\mu_p)]^T$ and $\mu_i$ stands for the $i-$th row of $\mu$.

## B  DATA GENERATION AND TRAINING STRATEGIES

### B.1  EXAMPLE 1 – INCOMPRESSIBLE NAVIER-STOKES EQUATION

For the two case study datasets in the incompressible Navier-Stokes example, we generate a total of 1,200 samples using the pseudo-spectral Crank-Nicolson solver available in Li et al. (2020c). We then split the generated dataset into 1,000, 100 and 100 for training, validation and testing, respectively. A histogram demonstration of the dataset distributions in small, medium and large data regimes is illustrated in Figure 3, where the test set of the small dataset of ntrain=10 samples exhibits a wider data distribution compared to its training dataset, which is aimed to test the data efficiency and out-of-distribution performances of the learned models. The fluid viscosity employed in the physics solver is $\nu = 10^{-4}$, and the timestep size is $\Delta t = 10^{-4}$ s. The solutions are obtained on a $256 \times 256$ spatial grid and the total duration of the simulation is 24 s. The obtained solutions are then downsampled to a $64 \times 64 \times 30$ grid, with the 3rd dimension being the temporal dimension. Note that, in downsampling the spatial dimensions, we employ a $2 \times 2$ mean pooling. This strategy is suggested in Helwig et al. (2023) to mitigate the spurious numerical errors not existed in the original data.

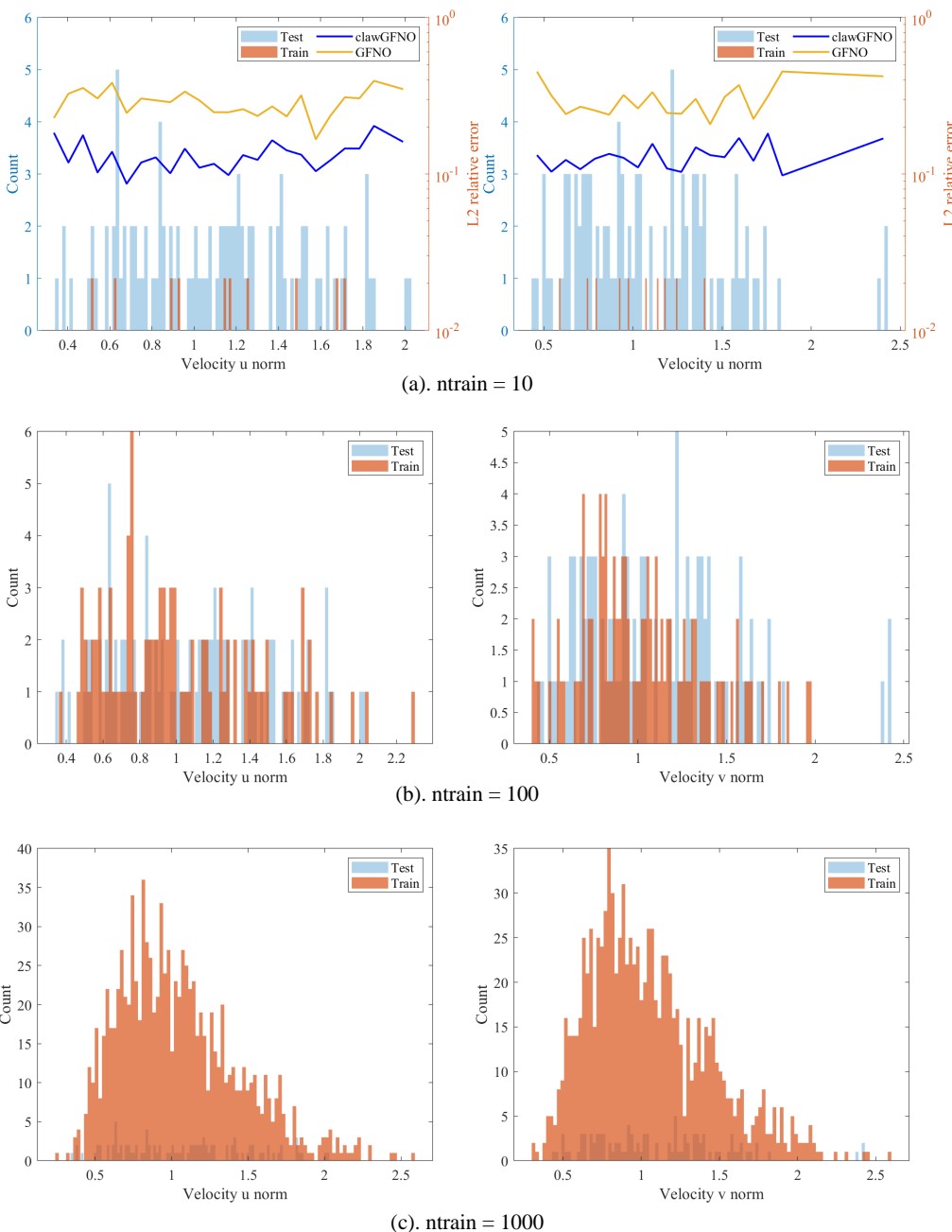

(a). ntrain = 10

(b). ntrain = 100

(c). ntrain = 1000

Figure 3: The data distribution of velocity in L2 norm, in the incompressible Navier-Stokes dataset. Left: the $x$ component of velocity. Right: the $y$ component of the velocity. Cases (a)-(c) represent the histogram of sample distributions in the small, medium and large data regimes, respectively, with blue representing the test samples and orange for the training samples. The per-sample relative L2 error on the test set is also plotted in (a), comparing clawGFNO (in navy) with GFNL (in yellow). This result demonstrates the improved the accuracy of clawNO, comparing to its counterpart, in small data regime.

To provide more details on the model performance, we plot the per-time-step prediction errors in terms of L2 relative error on the test dataset in Figure 4 using the best models trained with ntrain=1000 samples. Perhaps unsurprisingly, the prediction error increases as the prediction time step grows, due to the accumulation of error. All models have a similar growth rate, while clawNOs,

together with FNO, significantly outperform other baselines in accuracy. We also list a number of performance metrics in Table 5, including the total number of model parameters, the per-epoch runtime, the inference time, as well as the peak GPU usage. To quantitatively evaluate the divergence of the predicted solutions, we compute the averaged L2-norm of the divergence on the test dataset for all the models trained with ntrain=1000 samples (cf. the last row of Table 5). Because the additional layer in clawNOs are with pre-calculated weights, it barely adds any extra burden into GPU memory compared with its NO base model. Similar observation also applies to the runtime in large data regime. The inference time has a minor increase, due to the fact that the inferred solution will go through the an additional layer which avoidably increases the computational cost. When comparing the divergence of predicted solutions, we can see that both clawNOs predictions have much smaller divergence compared with all baselines. However, we point out that the solution divergence from clawNOs is not exactly zero, due to the numerical errors as discussed in Theorem 3.1.

**Remark:** The complexity of our clawNOs is very simular to their NO counterparts. Taking clawFNOs for example, the trainable part of the clawNO model consists of two fixed-size MLPs for the lifting layer and the projection layer, and $L$ numbers of Fourier layers between them. Denote $d_u$ as the input function dimension, $H$ as the latent dimension after lifting, $M$ as the total number of grids, $m$ as the number of Fourier modes on each dimension after truncation (which is often taken as half of the number of grids in each dimension, $M^{1/p}$, in practice), and $p$ as the problem dimension. During the lifting layer a vector valued input function taking values in $\mathbb{R}^{d_u}$ is linearly and locally mapped to the first layer feature function $\mathbf{h}(\cdot, 0)$ taking values in $\mathbb{R}^H$, and hence the number of trainable parameters is $Hd_u + H$. Then, each iterative Fourier layers involves the integral with a trainable kernel weight matrix in the Fourier domain, which is of size $2H^2m^p = 2^{1-p}H^2M$, and a local linear transformation which involves $H^2 + H$ numbers of trainable parameters. Then, the last iterative layer feature function, $\mathbf{h}(\cdot, L)$, is projected to the skew symmetric matrix-valued function $\mu$ with a two-layer MLP. Since the skew symmetric matrix-valued function $\mu$ is of degree of freedom $p(p-1)/2$ at each point $\mathbf{x}$, the projection layer maps a size $H$ input vector to a size $p(p-1)/2$ vector. Assume that the hidden layer of this MLP is of $d_Q$ neurons, the total number of trainable parameters in projection layer will be $Hd_Q + d_Q + d_Qp(p-1)/2 + p(p-1)/2$. Finally, $\mu$ will go through the pre-calculated differentiation layer, with $p^2(p-1)M^2/2 = p^2(p-1)m^{2p}/2$ numbers of non-trainable parameters in the FNO case. From the above calculation, we can see that clawFNO involves $(d_Q + H(d_u + d_Q + L + 1) + LH^2) + \dfrac{(d_Q + 1)}{2}p(p-1) + 2LH^2m^p$ numbers of trainable parameters, while the vanilla FNO involves $(d_Q + H(d_u + d_Q + L + 1) + LH^2) + (d_Q + 1)p + 2LH^2m^p$ numbers of trainable parameters. Therefore, the number of trainable parameters in clawFNO and FNO only differs in the second part of their projection layer, where clawFNO has $\dfrac{(d_Q + 1)}{2}p(p-1)$ numbers of parameters and FNO has $(d_Q + 1)p$. When $p > 3$, clawFNO will have a larger number of trainable parameters. However, we want to point out that since the number of parameter in the iterative layer $(2LH^2m^p)$ grows exponentially with dimension $p$, it dominates the cost, and the differences between clawFNO and FNO are negligible. This is consistent with what we observed in Table 5: the number of trainable parameters and the GPU cost of clawNOs and their counterparts are almost the same. During the inference, the non-trainable parameters in clawNO will play a role and we therefore observe an increase in the inference time.

Table 5: Performance comparison of selected models in incompressible Navier-Stokes case 1, in terms of the total number of parameters, the per-epoch runtime, inference time, peak GPU usage, and the L2-norm divergence of prediction. The runtime is evaluated on a single NVIDIA V100 GPU. Note that the case of ntrain=10 requiring more time to run compared to ntrain=100 is due to the reduced batch size of 2, as opposed to the batch size of 20 in ntrain=100.

| Case | ntrain | clawGFNO | clawFNO | GFNO | FNO | UNet | LSM | UNO | KNO |
|---|---|---|---|---|---|---|---|---|---|
| nparam (M) | | 0.85 | 0.93 | 0.85 | 0.93 | 0.92 | 1.21 | 1.07 | 0.89 |
| runtime (s) | 10 | 7.10 | 4.76 | 6.14 | 3.80 | 4.85 | 9.86 | 4.24 | 4.31 |
| | 100 | 4.89 | 2.42 | 4.39 | 2.03 | 2.39 | 4.98 | 2.69 | 2.35 |
| | 1000 | 41.75 | 19.56 | 40.06 | 17.38 | 20.80 | 37.86 | 18.78 | 18.20 |
| inf. time (s) | | 0.077 | 0.072 | 0.062 | 0.058 | 0.075 | 0.183 | 0.066 | 0.045 |
| GPU (GB) | 1000 | 0.68 | 0.34 | 0.68 | 0.34 | 0.12 | 13.20 | 2.14 | 0.16 |
| L2(div) | 1000 | 1.2e-3 | 3.8e-4 | 6.6e-2 | 4.7e-2 | 3.5e-1 | 5.4e-1 | 5.8e-1 | 1.8e-1 |

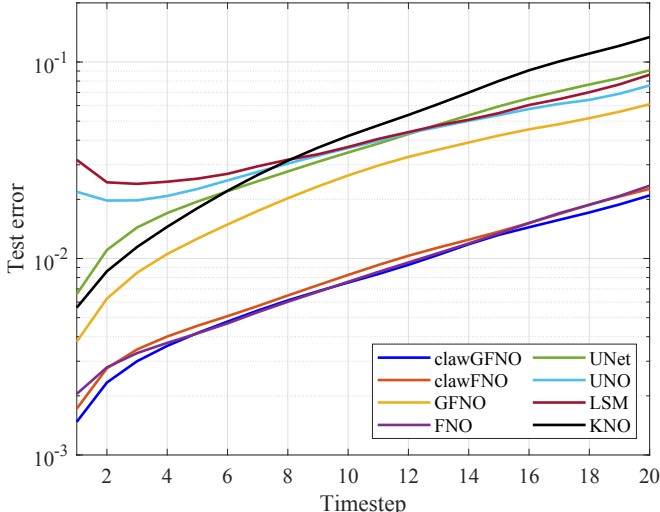

Figure 4: The per-time-step prediction error on the test dataset of the incompressible Navier-Stokes case 1, with ntrain=1000 samples.

## B.2 EXAMPLE 2 – RADIAL DAM BREAK

In generating the radial dam break dataset, we closely follow the numerical procedure in Takamoto et al. (2022), where we slightly modified the code to output the velocity fields in addition to the water height. A total of 1,000 samples are generated and are subsequently split into $n_{train}$, 100, 100 and training, validation, and testing, respectively, with $n_{train}$ the size of the training dataset depending on the adopted data regime. We run the numerical simulation on a $128 \times 128 \times 100$ grid and downsample the obtained solution to $32 \times 32 \times 25$ for training, where the first two dimensions are the spatial dimensions and the last is the temporal dimension. Analogous to the incompressible NS dataset, we perform $2 \times 2$ mean pooling in downsampling the spatial dimensions to maintain symmetry in data.

## B.3 EXAMPLE 3 – ATMOSPHERIC MODELING

As SpeedyWeather.jl uses spherical harmonics to solve the shallow water equations, we set the initial conditions $\eta_0$ for the aforementioned random waves through random coefficients of the spherical harmonics. The spherical harmonics are denoted as $Y_{\ell,m}$ with degree $\ell \geq 0$ and order $m$ with $-\ell \leq m \leq \ell$. Using a standard complex normal distribution $\mathcal{CN}(0,1) = \mathcal{N}(0,\frac{1}{2}) + i\mathcal{N}(0,\frac{1}{2})$, the random coefficients $\eta_{\ell,m}$ are drawn for degrees $10 \leq \ell < 20$ from $\mathcal{CN}(0,1)$, but $\eta_{\ell,0} \sim \mathcal{N}(0,1)$ for the zonal modes $m = 0$, and $\eta_{\ell,m} = 0$ otherwise. The wave lengths are $2\pi R\ell^{-1}$, about 2000 to 4000 km. The initial $u_0, v_0, \eta_0$ on a grid can be obtained through the spherical harmonic transform as

$$u_0 = v_0 = 0,$$
$$\eta_0 = A \sum_{\ell=0}^{\ell_{max}} \sum_{m=-\ell}^{\ell} \eta_{\ell,m} Y_{\ell,m}. \tag{14}$$

The amplitude $A$ is chosen so that $\max(|\eta_0|) = 2000$ m. The resolution of the simulation is determined by the largest resolved degree $\ell_{max}$, we use $\ell_{max} = 63$. In numerical weather prediction this spectral truncation is widely denoted as T63. We combine this spectral resolution with a regular longitude-latitude grid of 192x95 grid points ($\Delta\lambda = \Delta\theta = 1.875°$, about 200 km at the Equator, no grid points on the poles), also called a full Clenshaw-Curtis grid because of the underlying quadrature rule in the Legendre transform (Hotta & Ujiie, 2018). Non-linear terms are calculated on the grid, while the linear terms are calculated in spectral space of the spherical harmonics, and the model transforms between both spaces on every time step. This is a widely adopted method in global numerical weather prediction models.

For numerical stability, an implicitly calculated horizontal diffusion term of the form $-\nu\nabla^8\zeta, -\nu\nabla^8\mathcal{D}$, is added to the vorticity and the divergence equation, respectively. The power-4 Laplacian is chosen to be very scale-selective such that energy is only removed at the highest wave numbers, keeping the simulated flow otherwise largely unaffected. In practice, we use a non-dimensional Laplace operator $\tilde{\nabla}^2 = R^2\nabla^2$, such that the diffusion coefficient becomes an inverse time scale of $\tau = 2.4$ hours. The shallow water equations, equation 12, do not have a forcing or drag term such that the horizontal diffusion is the only term through which the system loses energy over time. The shallow water equations are otherwise energy conserving.

SpeedyWeather.jl employs a RAW-filtered (Williams, 2011) Leapfrog-based time integration with a time step of $\Delta t = 15$ min at T63 resolution. At this time step, the CFL number $C = c_{ph}\Delta t(\Delta x)^{-1}$ with equatorial $\Delta x = 2\pi R\frac{\Delta\lambda}{360°}$ is typically between $C = 1$ and $C = 1.4$, given wave speeds $c_{ph} = \sqrt{gh}$ between 280 and 320 ms$^{-1}$. Thanks to a centred semi-implicit integration of the linear terms (Hoskins & Simmons, 1975), the simulation remains stable without aggressively dampening the gravity waves with larger time steps or with a backwards implicit scheme. The continuity equation with the centred semi-implicit leapfrog integration reads as (the RAW-filter is neglected)

$$\frac{\eta_{i+1} - \eta_{i-1}}{2\Delta t} = -\frac{1}{2}\nabla\cdot(\mathbf{u}_{i+1}h_{i+1}) - \frac{1}{2}\nabla\cdot(\mathbf{u}_{i-1}h_{i-1}) \tag{15}$$

with previous time step $i - 1$, and next time step $i + 1$. The RAW-filter then acts as a weakly dampening Laplacian in time, coupling the tendencies at $i-1, i$ and $i+1$ to prevent a computational mode from growing.

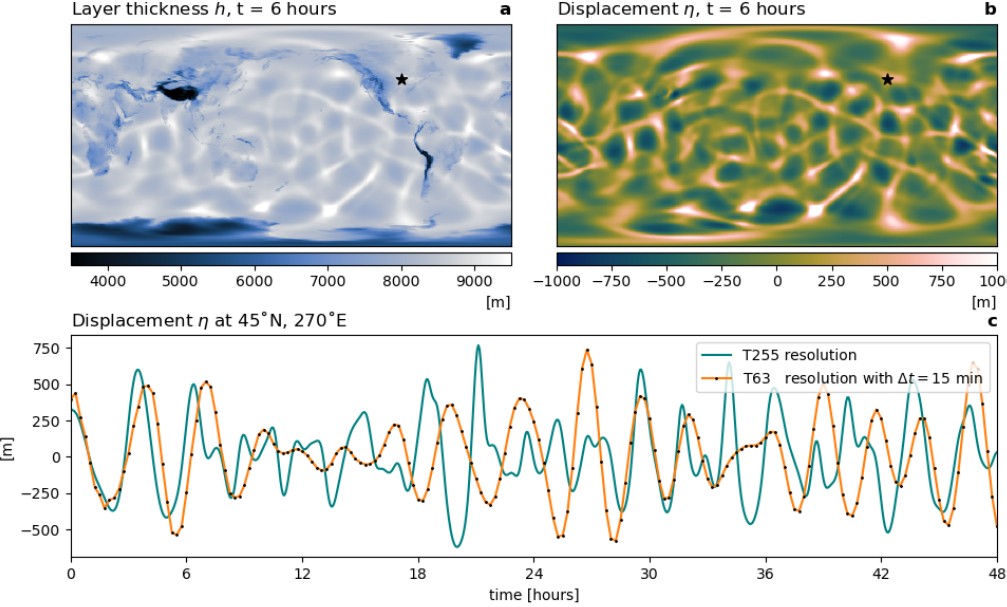

Figure 5: Atmospheric gravity waves as simulated by SpeedyWeather.jl. **a,** Layer thickness $h$ and **b** displacement $\eta$ after $t = 6$ hours at a resolution of T255 (about 50 km). The layer thickness $h$ includes, in contrast to $\eta$, clearly the signal of the underlying orography of Earth. **c** Time series of $\eta$ over the USA (45°N, 90°W, marked with a black star in **a,b**). Black circle markers denote the time step $\Delta t = 15$ min used for model integration and training data. Both simulations, T63 and T255, started from the same initial conditions, illustrating the limited predictability of the shallow water equations.

While the initial conditions contain only waves of wave lengths 2000 to 4000 km shorter waves are created during the simulation due to non-linear wave-wave interactions and interactions with the Earth's orography (Fig. 5). The time scale of these gravity waves is on the order of hours (Fig. 5c), which is why we use a training data sampling time step of $\Delta t = 15$ min. Much longer time steps as used by Gupta & Brandstetter (2022) would therefore fail to capture the wave dynamics present in the shallow water simulations. It is possible to use initial conditions that are closer to geostrophy, such that the presence of gravity waves from geostrophically-unbalanced initial conditions is

reduced. In such a setup, the predictability horizon is given by the chaotic vorticity advection and turbulence that evolves over longer time scales, which would justify a longer data sampling time step. However, the Gupta & Brandstetter (2022) setup includes a strong gravity wave in the initial conditions that propagates meridionally, while also including some slowly evolving vortices. Our setup therefore represents a physically clearer defined problem, one that focuses on the non-linear gravity wave propagation in the shallow water system. Following equation 14, our setup can be easily recreated in other models for further studies. In this context, we generate a total of 1,200 samples and split them into 1,000/100/100 for training/validation/testing, respectively.

### B.4 EXAMPLE 4 – CONSTITUTIVE MODELING OF MATERIAL DEFORMATION

We generate the material deformation dataset as follows. Firstly, we generate a solution $A$ from a steady-state 2D Darcy flow with the diffusion coefficient specified as a random Gaussian field, following the numerical procedure in Li et al. (2020c). The ground truth $\mathbf{u}$ is generated as

$$\mathbf{u} = (\frac{\partial A}{\partial y} , - \frac{\partial A}{\partial x}) ,$$

which is guaranteed divergence-free, and the body load $\mathbf{f}$ is then computed following equation 13. The derivatives are numerically computed using FFT on a regular mesh. A total of 300 samples are generated and are subsequently split into $n_{train}$, 100, 100 for training, validation, and testing, respectively, with $n_{train}$ the size of the training dataset depending on the adopted data regime. We run the numerical simulation on a $80 \times 80$ uniform mesh and downsample the solution to a grid of 246 spatial points through interpolation on a circular domain of radius $0.4$ and centers at the origin point.

### B.5 TRAINING STRATEGIES

We run three replicates for all the experiments and report the mean and standard deviation of the L2 relative error for comparison metrics. For all Fourier-domain models, we closely follow the model setup in Helwig et al. (2023), employing four Fourier layers and keeping only the 12 lowest Fourier modes in all the 2D models and 8 spatial and 6 temporal lowest modes in all the 3D models. An exception is in the atmospheric modeling problem, where we truncate the spatial Fourier modes to 22 for correct physical realizability. For fair comparison, we adopt Cartesian encoding in all the models.

Similar to the model size in Helwig et al. (2023), we set the latent dimension in FNO and clawFNO to 20, whereas we counterbalance the additional dimensions introduced due to equivariance in GFNO and clawGFNO by reducing the latent dimension to 10 in all the 2D models and 11 in all the 3D models. For UNet, in order to arrive at a similar number of model parameters, we increase the first-layer dimension to 11 in the incompressible NS problem, to 15 in the radial dam break problem, and to 96 in weather modeling.

As suggested in Tran et al. (2022) and Helwig et al. (2023), we turn to the teacher forcing strategy to facilitate the learning process. We set the batch size to 20 for all 2D models and 10 for all 3D models, with the exception in small and medium data regimes, where we set batch size to 2 and 1 when the training datasets are of size 10 and 2, respectively. We employ cosine annealing learning rate scheduler that decays the initial learning rate to 0. All the 2D models are trained for a total of 100 epochs whereas all the 3D models are trained for 500 epochs with an early stop if the validation loss stops improving for consecutive 100 epochs. 2D models are trained with less number of epochs as one training sample in 3D corresponds to $(T - T_{in})$ training samples in 2D. We directly take the baseline models in GFNO (Helwig et al., 2023), and further tune the hyperparameters (i.e., the learning rate and weight decay in the Adam optimizer) in clawNOs. All the experiments are carried out on a single NVIDIA A6000 40GB GPU.

For both of the two graph-based models (i.e., INO and GNO), we closely follow the model setup in Liu et al. (2023) and Li et al. (2020c). In order to be consistent across clawINO, INO and GNO, the latent width is set to 64 and the kernel width is set to 1,024 in all models. We employ a total of 4 integral layers in all models, whereas the shallow-to-deep technique is equipped for initialization in INO-based models. The batch size is set to 2 for all the irregular-mesh models, with the exception in the first case in the constitutive modeling example where a batch size equal to 1 is employed.

We adopt the cosine annealing learning rate scheduler that decays the initial learning rate to 0. All the models are trained for 500 epochs with an early stop if the validation loss stops improving for consecutive 200 epochs.

## C   ROLLOUT VISUALIZATIONS

We illustrate the rollout of randomly selected trajectories in the test dataset using clawNO predictions, along with the comparisons against ground truth data and the corresponding absolute errors. The rollouts of incompressible NS, radial dam break, and atmospheric modeling are showcased in Figure 6, Figures 8 and 9, Figures 11 and 2, respectively. We also showcase the material deformation prediction in 6 different test samples in Figure 15. To provide a visual comparison across models, we plot the final-step prediction of all the models in the incompressible NS example in Figure 7, the radial dam break example in Figure 10, and the atmospheric modeling example in Figure 14.

## D   DETAILED DERIVATIONS

### D.1   PROOF OF THEOREM 3.1

Since the Fourier spectral differentiation error estimate is a direct result of Trefethen (2000, Page 34), we provide the detailed derivation of equation 7 in this section. According to the basic properties of Fourier transform, if $f$ is a differential and periodic function on $[0, L]$ with its Fourier representation:

$$f(\mathbf{x}) = \sum_{\xi=-N/2}^{N/2-1} \hat{f}(\xi) e^{i2\pi\xi/L},$$

its derivative can be given as

$$f'(\mathbf{x}) = \sum_{\xi=-N/2}^{N/2-1} \frac{i2\pi\xi}{L} \hat{f}(\xi) e^{i2\pi\xi/L} \approx \mathcal{F}^{-1} \left[ \frac{i2\pi\xi}{L} \mathcal{F}[f](\xi) \right].$$

As such, equation 7 can be obtained by applying the above property to approximate every derivative term $\dfrac{\partial \mu_{jk}^{(N^{(1)}, \cdots, N^{(p)})}}{\partial x_k}$.

### D.2   PROOF OF THEOREM 3.2

It suffices to show that $\left| \dfrac{\partial \psi}{\partial x_k}(\mathbf{x}_i) - \sum_{\mathbf{x}_l \in \chi \cap B_\delta(\mathbf{x}_i)} (\psi(\mathbf{x}_l) - \psi(\mathbf{x}_i)) \omega_{i,l}^{(k)} \right| \leq C\Delta x^{m+1}$ for any $\psi \in$

$C^{m+1}(\Omega)$. Denote $I[\psi](\mathbf{x}_i) := \dfrac{\partial \psi}{\partial x_k}(\mathbf{x}_i)$, $I_{\Delta x}[\psi](\mathbf{x}_i) := \sum_{\mathbf{x}_l \in \chi \cap B_\delta(\mathbf{x}_i)} (\psi(\mathbf{x}_l) - \psi(\mathbf{x}_i)) \omega_{i,l}^{(k)}$ and let $\phi_m$ denote the $m-$th order truncated Taylor series of $\psi$ about $\mathbf{x}_i$ with associated remainder $r_m$, such that

$$\psi(\mathbf{y}) = \phi_m(\mathbf{y}) + r_m(\mathbf{y}) = \sum_{|\alpha| \leq m} \frac{D^\alpha \psi(\mathbf{x}_i)}{\alpha!} (\mathbf{y} - \mathbf{x}_i)^\alpha + \sum_{|\beta|=m+1} R_\beta(\mathbf{y}) (\mathbf{y} - \mathbf{x}_i)^\beta,$$

where $R_\beta(\mathbf{y}) := \dfrac{|\beta|}{\beta!} \int_0^1 (1 - \tau)^{|\beta|-1} D^\beta u(\mathbf{y} + \tau(\mathbf{y} - \mathbf{x}_i)) d\tau$ and therefore $|R_\beta(\mathbf{y})| \leq$

$\dfrac{1}{\beta!} \max_{|\alpha|=m+1} \max_{\mathbf{y} \in B_\delta(\mathbf{x}_i)} |D^\alpha \psi(\mathbf{y})| \leq C \|\psi\|_{C^{m+1}}$. We then have

$$|\psi - \phi_m|(\mathbf{y}) = |r_m|(\mathbf{y}) = \left| \sum_{|\beta|=m+1} R_\beta(\mathbf{y}) (\mathbf{y} - \mathbf{x}_i)^\beta \right|$$

$$\leq |\mathbf{y} - \mathbf{x}_i|^{m+1} \sum_{|\beta|=m+1} |R_\beta(\mathbf{y})| \leq C \|\psi\|_{C^{m+1}} |\mathbf{y} - \mathbf{x}_i|^{m+1}.$$

To bound the approximation error, we apply the triangle inequality and the reproducing condition of polynomial $\phi_m$:

$$\begin{aligned}
|I[\psi](\mathbf{x}_i) - I_{\Delta x}[\psi](\mathbf{x}_i)| \leq & |I[\psi](\mathbf{x}_i) - I[\phi_m](\mathbf{x}_i)| + |I[\phi_m](\mathbf{x}_i) - I_{\Delta x}[\psi](\mathbf{x}_i)| \\
= & |I[\psi](\mathbf{x}_i) - I[\phi_m](\mathbf{x}_i)| + |I_{\Delta x}[\phi_m](\mathbf{x}_i) - I_{\Delta x}[\psi](\mathbf{x}_i)|.
\end{aligned}$$

Here, the first term vanishes since $\phi_m$ is the truncated Taylor series of $\psi$ and $m \geq 1$, and for the second term we have

$$\begin{aligned}
|I_{\Delta x}[\phi_m](\mathbf{x}_i) - I_{\Delta x}[\psi](\mathbf{x}_i)| \leq & \sum_{\mathbf{x}_l \in \chi \cap B_\delta(\mathbf{x}_i)} |\psi(\mathbf{x}_l) - \psi(\mathbf{x}_i) - \phi_m(\mathbf{x}_l) + \phi_m(\mathbf{x}_i)||\omega_{i,l}^{(k)}| \\
= & \sum_{\mathbf{x}_l \in \chi \cap B_\delta(\mathbf{x}_i)} |\psi(\mathbf{x}_l) - \phi_m(\mathbf{x}_l)||\omega_{i,l}^{(k)}| \\
\leq & C||\psi||_{C^{m+1}} \sum_{\mathbf{x}_l \in \chi \cap B_\delta(\mathbf{x}_i)} |\mathbf{x}_l - \mathbf{x}_i|^{m+1}|\omega_{i,l}^{(k)}| \\
\leq & C||\psi||_{C^{m+1}} \Delta x^{m+1} \sum_{\mathbf{x}_l \in \chi \cap B_\delta(\mathbf{x}_i)} |\omega_{i,l}^{(k)}| \leq C\Delta x^{m+1}.
\end{aligned}$$

Here, the last inequality can be proved following the argument in Levin (1998, Theorem 5): for each fixed $k$ and $i$, the coefficient $\omega_{i,l}^{(k)}$ is a continuous function of $\mathbf{x}_l$. Moreover, the size of $\chi \cap B_\delta(\mathbf{x})$ is bounded. Thus, for a fixed $\Delta x$ it follows $\sum_{\mathbf{x}_l \in \chi \cap B_\delta(\mathbf{x}_i)} |\omega_{i,l}^{(k)}| \leq C$ where $C$ is independent of $\Delta x$, $k$, and $\mathbf{x}_i$. And therefore we obtain $|I[\psi](\mathbf{x}_i) - I_{\Delta x}[\psi](\mathbf{x}_i)| \leq C\Delta x^{m+1}$ and finish the proof.

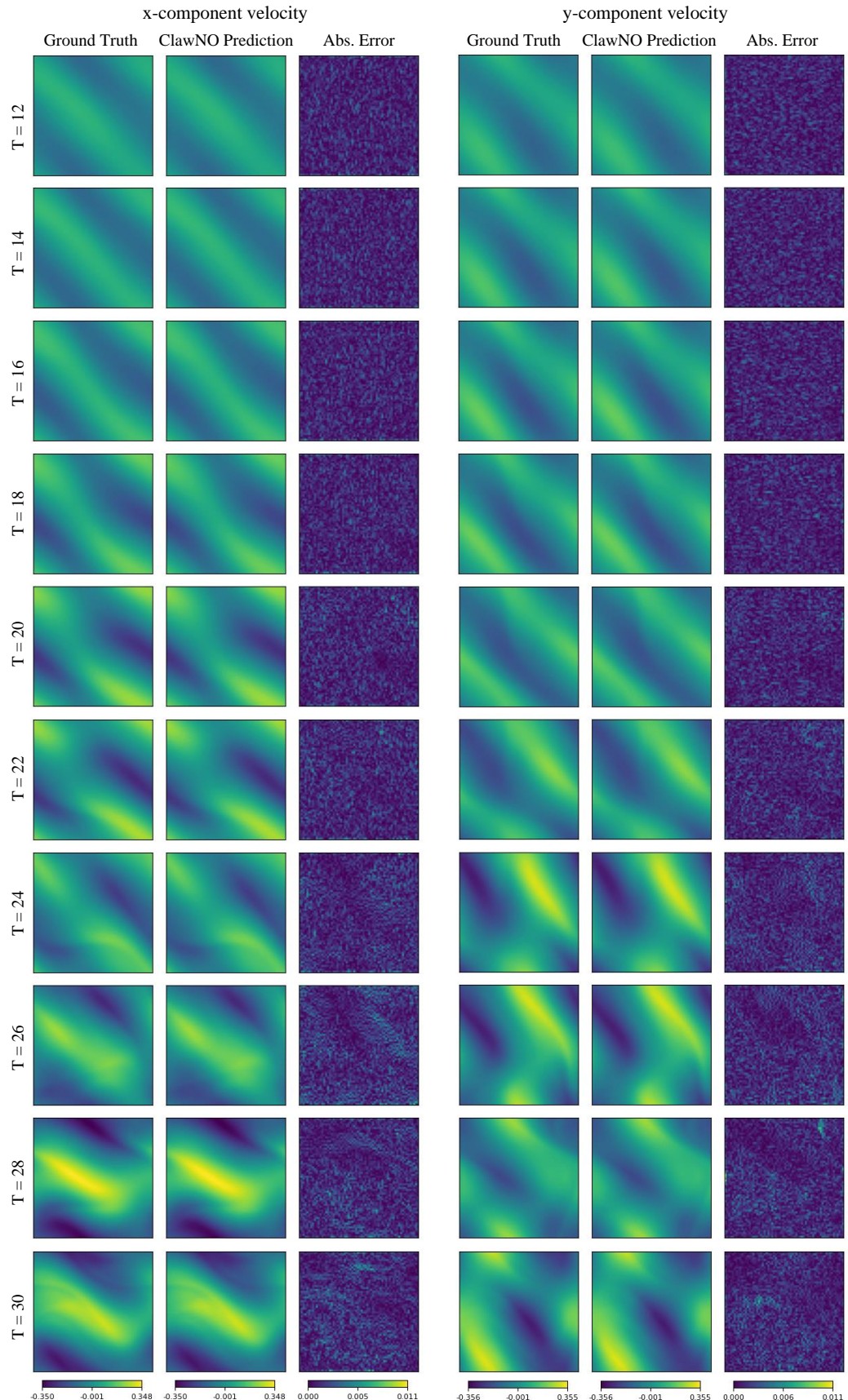

Figure 6: Rollout of incompressible Navier-Stokes case 1.

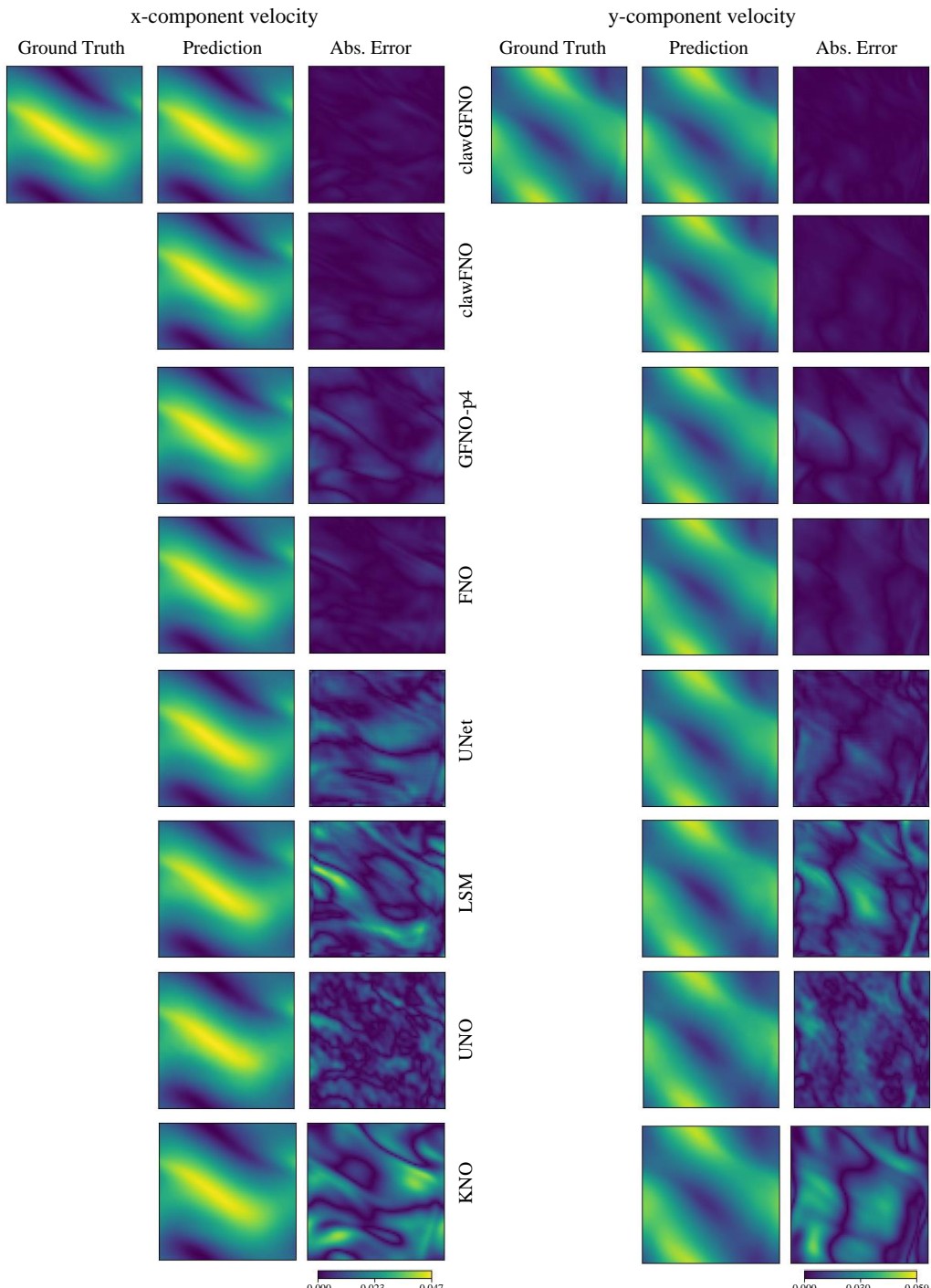

Figure 7: Last-step prediction comparison across models in incompressible Navier-Stokes case 1.

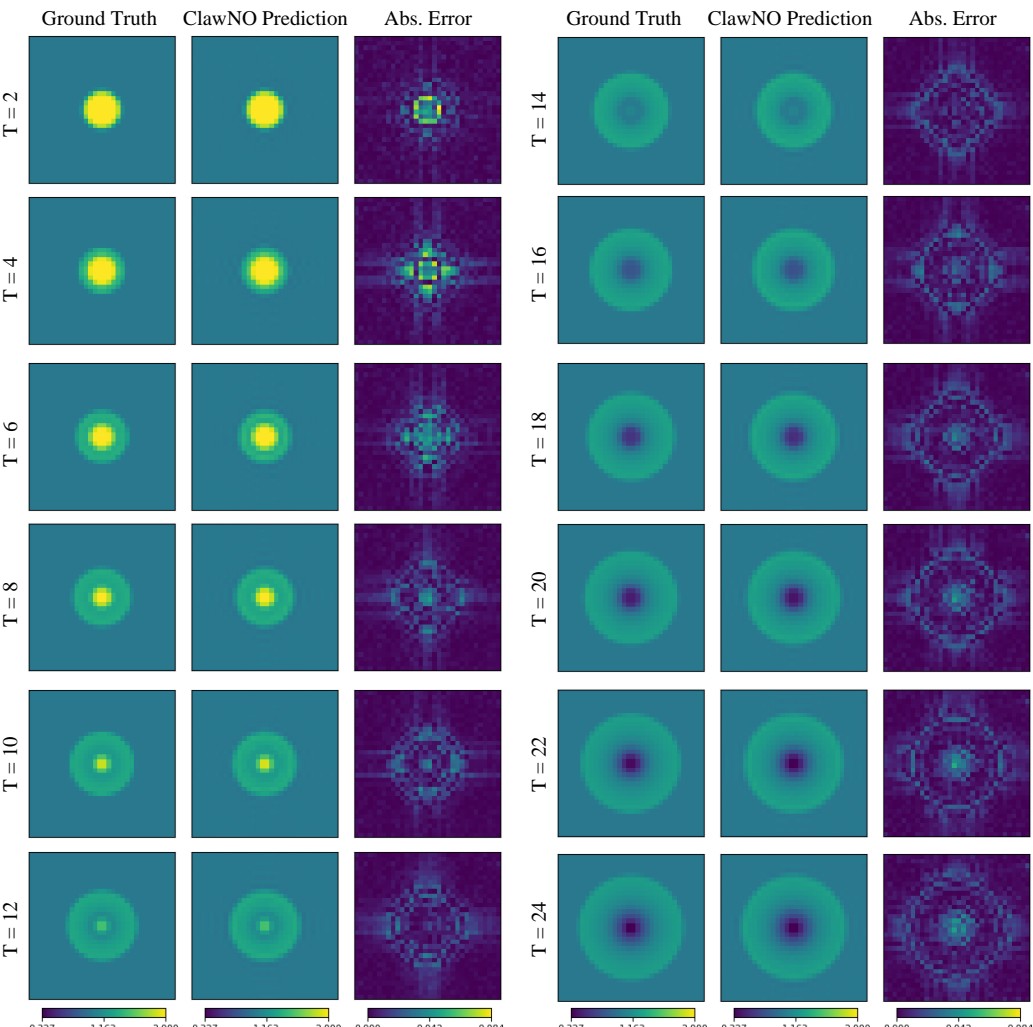

Figure 8: Rollout of water depth in radial dam break modeling.

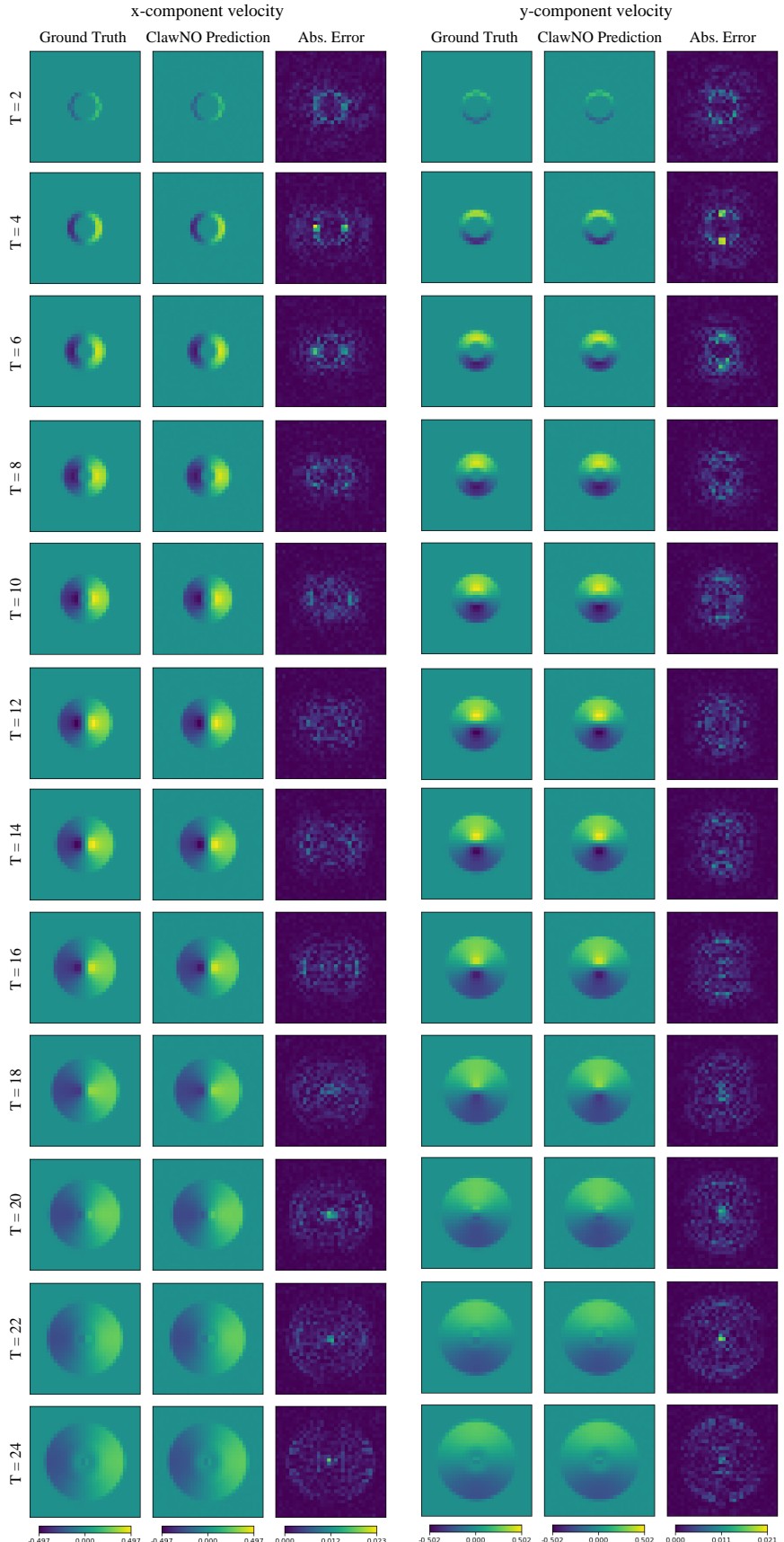

Figure 9: Rollout of velocity in radial dam break modeling.

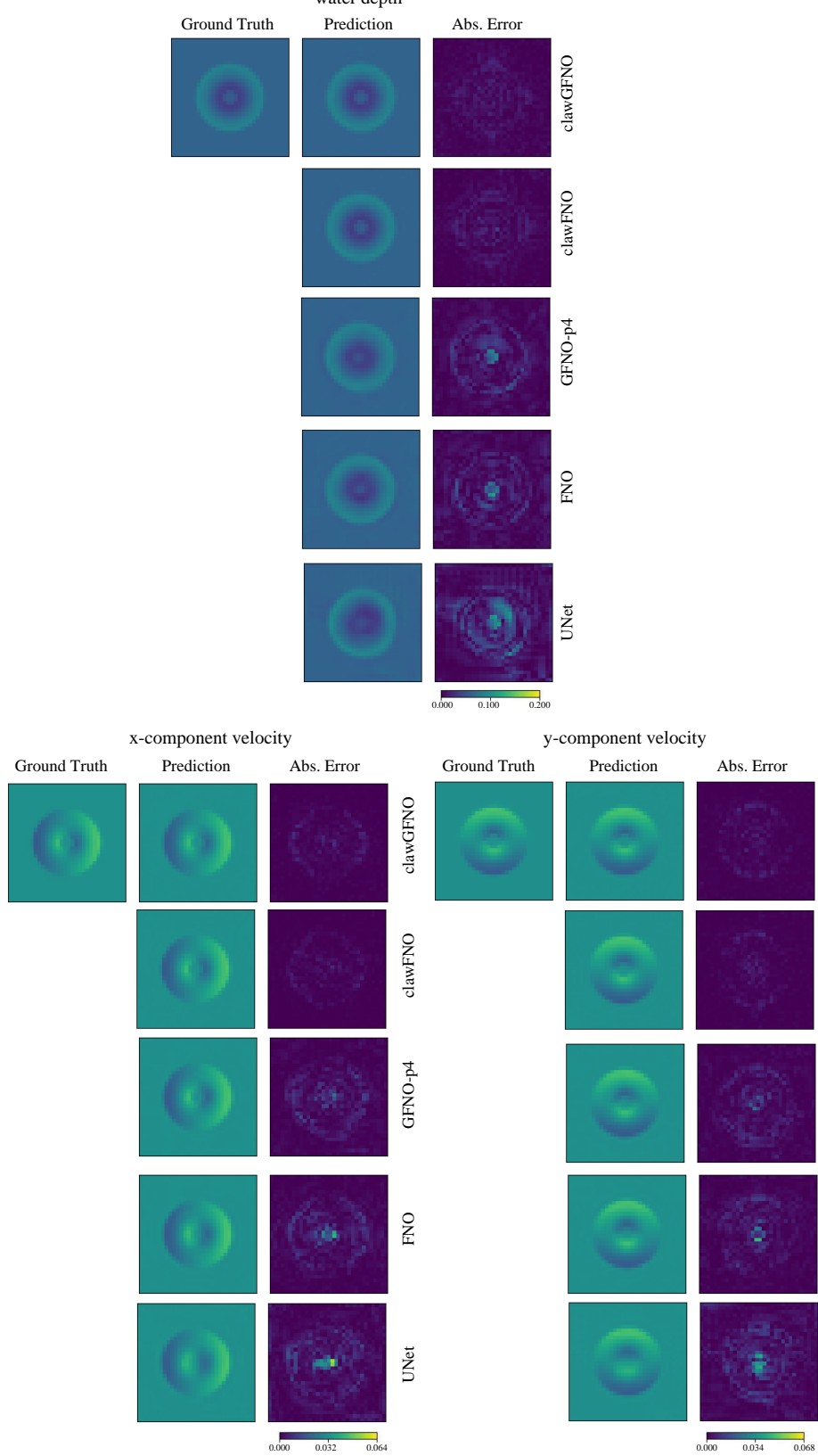

Figure 10: Last-step prediction comparison across models trained with ntrain=10 samples in radial dam break dataset.

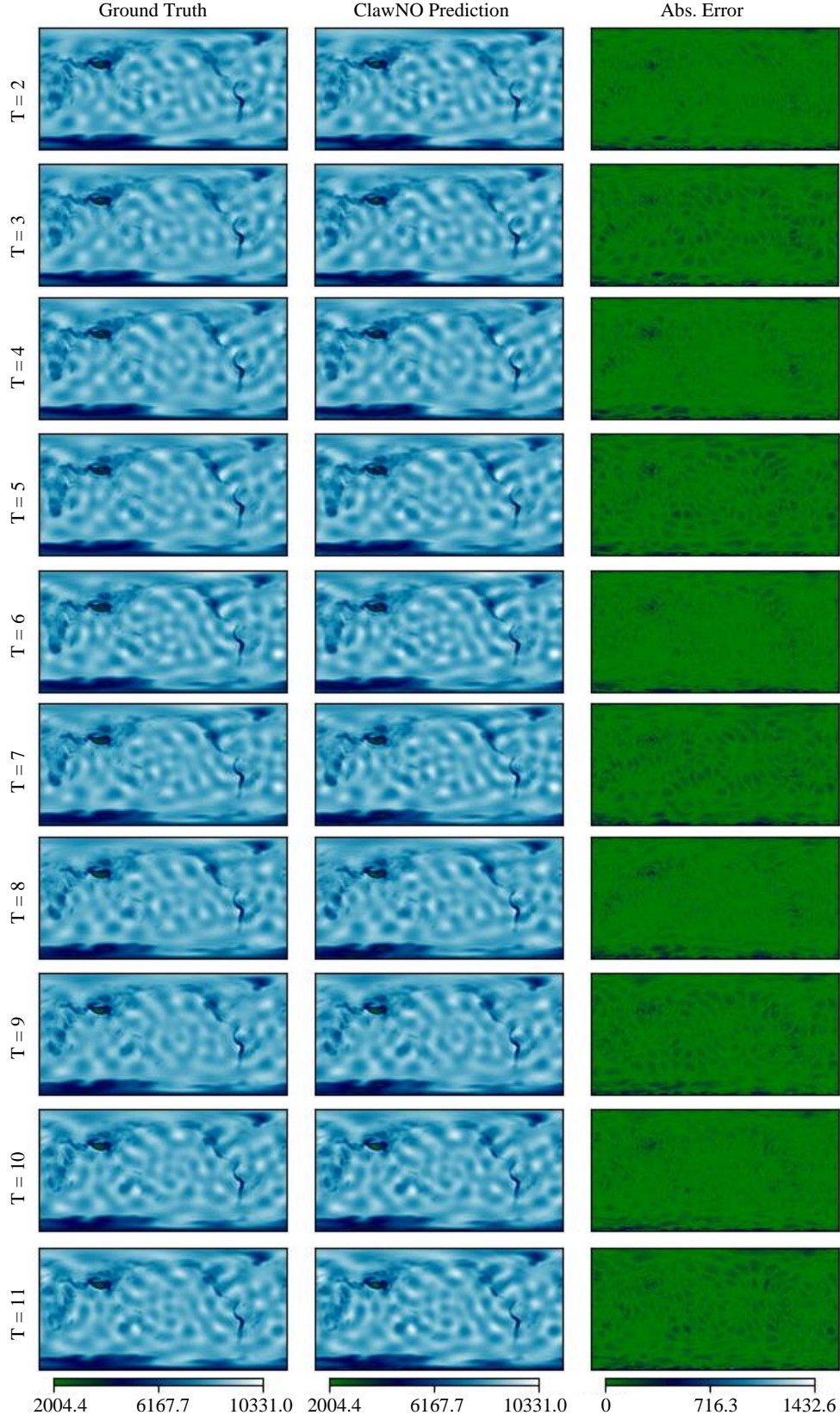

Figure 11: Rollout of layer thickness in atmospheric modeling.

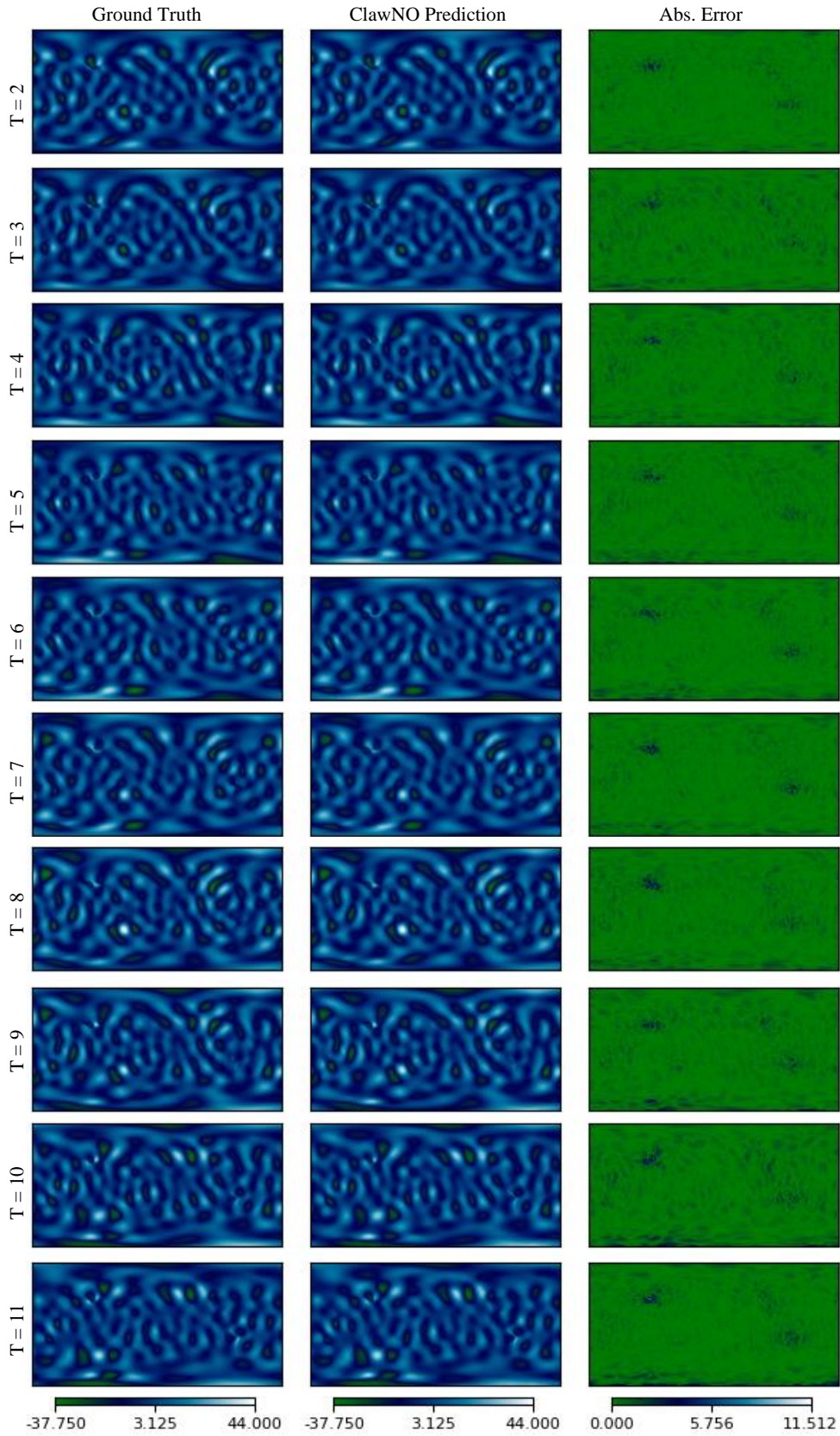

Figure 12: Rollout of zonal wind velocity in atmospheric modeling.

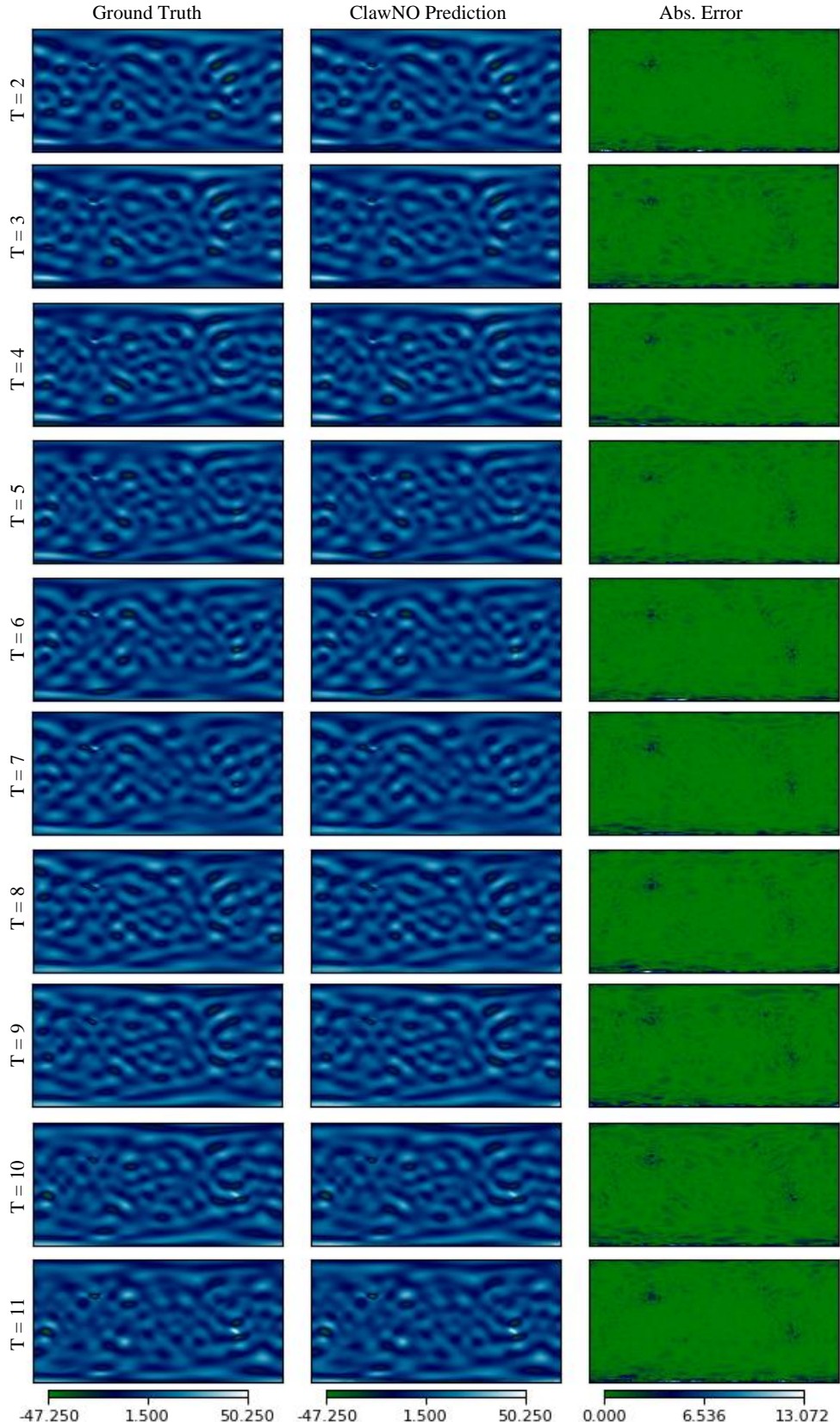

Figure 13: Rollout of meridional wind velocity in atmospheric modeling.

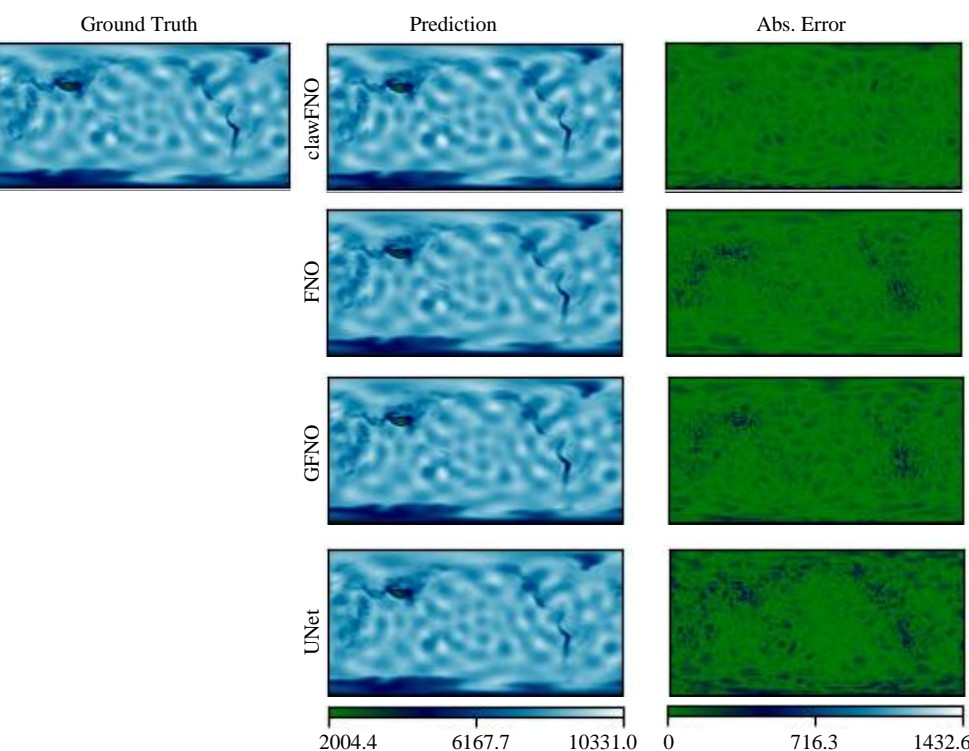

Figure 14: Last-step prediction comparison across models in atmospheric modeling dataset.

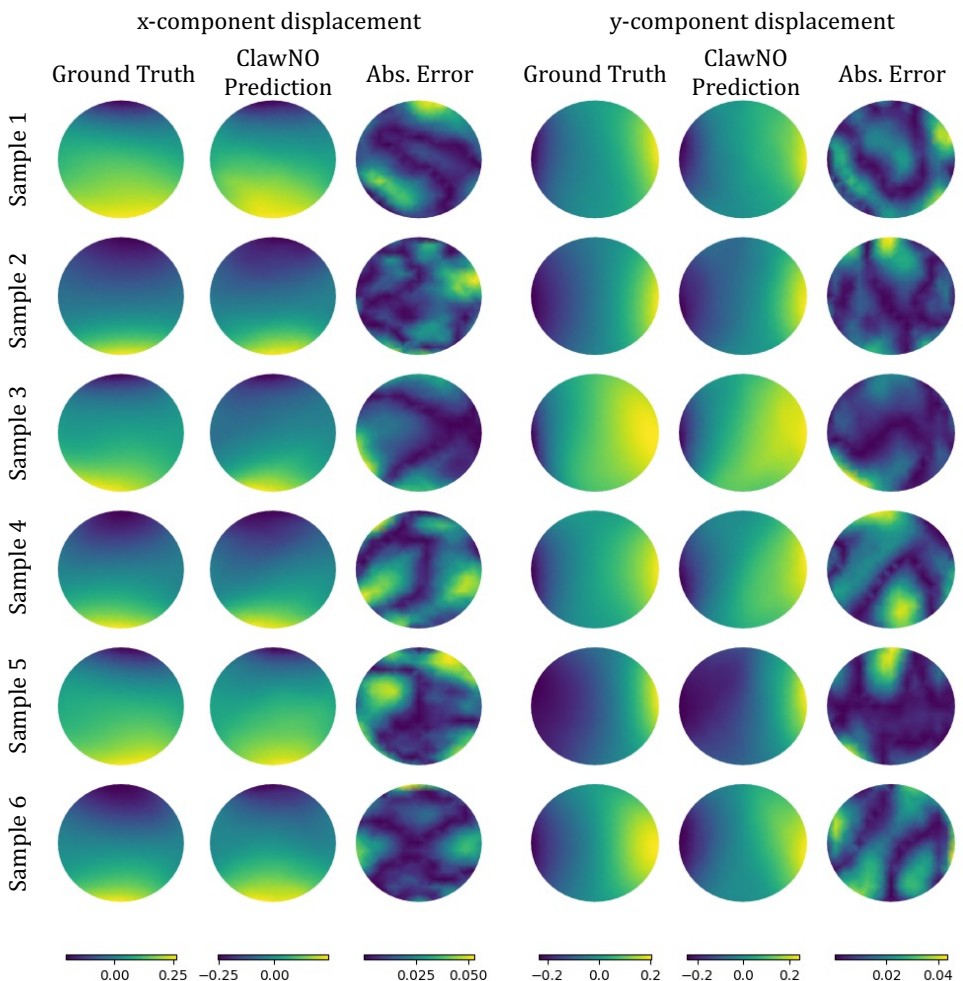

Figure 15: Demonstration of the constitutive modeling of material deformation.

