# OpenReview forum: "Harnessing the Power of Neural Operators with Automatically Encoded Conservation Laws"
_ICLR.cc/2024/Conference — Submitted to ICLR 2024_

### Official Review · Reviewer_L28W · 2023-10-24

**Soundness:** 2 fair
**Presentation:** 2 fair
**Contribution:** 2 fair
**Rating:** 5
**Confidence:** 4

**Summary:**

This paper presents the clawNOs to automatically encode the conservation law to neural operators. Instead of directly learning final solution, clawNOs propose to learn a substitute skew-symmetric matrix-valued function, whose differentiation naturally holds the divergence-free property. Technologically, a special differentiation layer is proposed for implementation. clawNOs surpass FNO and geo-FNO in three benchmarks and performs well in small-data regimes.

**Strengths:**

-	The proposed method performs well in three typical benchmarks.

-	Encoding the conservation law can boost the model performance in small-data regimes.

**Weaknesses:**

1. About the novelty.

The analysis in Section 3.1 is similar to this following paper. But they didn't cite this paper in this section, making it is hard to judge their contribution.

Neural Conservation Laws: A Divergence-Free Perspective, NeurIPS 2022.

2. About the efficiency.

Since clawNOs introduce an additional differentiation layer, the comparisons w.r.t. FNO and GFNO in running time, GPU memory, model parameter are expected.

3.	More baselines.

Actually, FNO and GFNO are both simple baselines. Recently, there are many advanced neural operators, such as LSM [1], U-NO [2]. It is hard to judge the model performance unless they compare with the above mentioned baselines.

[1] Solving High-Dimensional PDEs with Latent Spectral Models, ICML 2023

[2] U-NO: U-shaped Neural Operators, TMLR 2023

4. More analysis about the divergence-free property of outputs.

I can understand that if the model does learn an optimal differentiation layer, the output results will strictly satisfy the divergence-free property. But the authors adopt an approximation design. How well will the differentiation layer perform? A metric for the output divergence-free property is expected. Does clawNOs really perform better than FNO or GFNO in learning divergence-free results? Experiments are required.

5. More showcase comparisons.
The authors only provide the showcases for their own model in the supplementary materials. More showcase comparisons are expected to intuitively demonstrate the advantages of clawNOs.

6. About the long-term prediction.
Since the authors claim the long-term performance as an important advancement of clawNOs, they should plot the time-stamp-wise error curves in comparison with other baselines. Concretely, how does the prediction error change along the temporal dimension?

7. The authors claim that “ClawNOs significantly outperform the state-of-the-art NOs in both accuracy and generalizability” in the abstract. What does the “generalizability” mean here?

8. I am also confused about the physical understanding of clawNOs. Are there any physical meanings to the skew-symmetric matrix-valued function $\mu$? Or this is just a mathematical trick. I think this question is important, since it determines whether clawNOs hold potential physical limitations or not.

**Questions:**

Since they do not properly cite the previous work, it is hard to judge their contribution. Besides, the experiments and clarifications about their design are insufficient. All the details are included in the weaknesses section.

---

> ### Author Response · Authors · 2023-11-20
> **Response, part 1**
>
> We thank the reviewer's valuable time and insightful suggestions, and have added new clarifications, new baselines, and new tests per the reviewer's suggestions. Our response:
>
> **Comparison with [1] and our novelty**: We would like to point out that we have cited [1] and discussed the differences and limitations in their setting, in the last paragraph of Section 2 in the original manuscript. Although our Section 3.1 has described the differential form representation as in [1], the novelty of our idea is NOT on formulating the conservation law into a divergence-free architecture with a differential form representation. In fact, such a representation is well-known for more than a decade, see, e.g., [2], way before [1]. Therefore, it is not the novelty of [1] either. We have added more references on differential forms in the revised manuscript, to avoid such a confusion.
>
> The novelty of our approach is 1) the combination of neural operator with the mass conservation as a basic physical law, to learn the unknown physical system responses such as the constitutive laws with improved data efficiency; and 2) the design of a pre-calculated numerical differentiation layer in the neural operator architecture, to enable the incorporation of differential form-guaranteed divergence-free output, without increasing the number of trainable parameters and computational cost. To our best knowledge, neither of these two ideas have been discussed in existing neural operator literature. In [1] and all other PINN-based architectures, they are designed based on the knowledge of full governing laws, while we (as the general neural operator papers) consider the hidden physics scenario where the full PDE is not given. In the PINNs setting, derivatives are evaluated via automatic differentiation and hence the evaluation of differential forms is trivial in [1]. However, in neural operators we need to design approaches to evaluate the derivatives. This is also the main focus of section 3.2.
>
>
> [1] Richter-Powell, Jack, Yaron Lipman, and Ricky TQ Chen. "Neural conservation laws: A divergence-free perspective." Advances in Neural Information Processing Systems 35 (2022): 38075-38088.
>
> [2] Barbarosie, Cristian. "Representation of divergence-free vector fields." Quarterly of applied mathematics 69.2 (2011): 309-316.
>
> **Efficiency**: We thank the reviewer's suggestion, and have added table 5 in the appendix of the revised manuscript for detailed performance comparison of selected models. The comparison metrics include the total number of model parameters, the per-epoch runtime, inference time, the peak GPU memory usage, and the L2-norm of the output divergence.
>
> Note that the weights in our additional differential layer is pre-calculated, and it therefore contains no trainable parameters. As shown in table 5 of the revised paper, the number of trainable parameters and the GPU cost of clawNOs and their counterparts are almost the same. During inference, although the number of parameters in iterative layers still dominate, the non-trainable parameters in clawNO will play a role. Therefore, we do observe a minor increase in inference time.
>
> **More baselines**: Following the reviewer's suggestion, we have added three state-of-the-art neural operator baselines, i.e., LSM, UNO, and KNO, in our incompressible Navier-Stokes experiment in table 1 of the revised manuscript. Our original conclusion still stands: clawNOs consistently outperform all baselines in all three considered data regimes.
>
> **Analysis of the output divergence**: The differentiation layer in clawNOs is not trained through data, but pre-calculated through either FFT or through pre-computed weights, so the divergence-free property of the output is guaranteed. To investigate the effect of this differentiation layer, we compute the averaged L2-norm divergence on the test dataset for all the models trained with ntrain=1000 samples, and report the results in table 5 in the appendix of the revised manuscript. As one can see in the table, the L2-norm divergence is close to zero for both clawGFNO and clawFNO, whereas other non-claw baselines have relative large divergence.
>
> **More showcase comparisons**: We thank the reviewer for their great suggestion. We have followed the LSM paper and included more showcase comparisons (Figs. 7, 10 and 14) in the revised paper, providing additional qualitative comparison between different models.
>
> **Time-stamp-wise error**: The time-stamp-wise error is added in Fig. 4 in the revised paper. As expected, the prediction error increases as the prediction time step grows, due to the accumulation of error. All models have a similar growth rate, while clawNOs, together with FNO, significantly outperform other baselines in accuracy.

---

> ### Author Response · Authors · 2023-11-20
> **Response, part 2**
>
> **Generalizability**: Our small data regime setting is designed to test the performance of the trained models when the available data is scarce and does not provide a good coverage of the parameter range (i.e., out-of-domain). Taking the incompressible Navier-Stokes experiment as an example, when ntrain is as small as $10$, it is expected that some test samples will be out of the training regime. To validate this, in the paper revision we show in Fig. 3 of the appendix the data distribution in the incompressible Navier-Stokes experiment case 1. In particular, the histogram of the solution L2-norms are plotted in the $x$ and $y$ components. One can see that in the small data regime, the training set does not provide a good coverage of the dataset, and a large number of test samples are outside the training regime. Then, to illustrate the generalizability of clawNOs, we also plot the per-sample test errors in Fig. 3a, where one can see that clawNO's error is much smaller than that of the non-claw counterpart, especially in the out-of-distribution regime. The training datasets of the medium and large data regimes provide good coverage of the entire dataset and aim to test the in-domain predictability of various models.
>
> **Physical meaning to the skew-symmetric matrix-valued function**: There is no unified physical meaning for the skew-symmetric matrix-valued function. Because the clawNOs have no restriction on the applicable physical system, the physical meaning of solution as well as $\mu$ depend on the specific problem. Taking the 2D Navier-Stokes equation as an example, the skew-symmetric matrix has only one degree of freedom, $\mu_{12}$, satisfying $\dfrac{\partial \mu_{12}}{\partial x}=u$ and $\dfrac{\partial \mu_{12}}{\partial y}=v$. Here, $u$ and $v$ are the $x-$ and $y-$ components of the velocity field, respectively. In this context,  $\mu_{12}$ is the potential function defined in classical hydrodynamics literature.
>
> Thank you once again for your excellent suggestions. We would be happy to answer any follow-up or additional questions you may have.

---

> > ### Comment · Reviewer_L28W · 2023-11-22
> > **Thanks for your response**
> >
> > Thanks for your effort in clarifying the differences from previous methods, and adding new baselines, showcases, and efficiency comparisons. Most of my concerns are resolved.
> >
> > But I think some of the key claims are not rigorous, such as the "long-term prediction", where the new results present that all models have a similar growth rate. Usually, some papers focusing on the accumulation error will highlight their long-term prediction capability. Thus, I insist that this is an unsuitable claim.
> >
> > Also about the "Generalizability", this description is usually used for different domain settings. I think that the authors mean "data efficiency" here.
> >
> > Thus, I would like to raise my score to 5. Thanks for the author's effort.

---

> > > ### Author Response · Authors · 2023-11-22
> > > **Thank you and further revision**
> > >
> > > We thank the reviewer for reading our rebuttal, and for kindly raising the score. We also appreciate the reviewer's further suggestions. We have revised the claims about generalizability and long-term prediction in a more precise way as the reviewer have suggested. All new edits are marked by blue.

---

### Official Review · Reviewer_t5ea · 2023-11-06

**Soundness:** 3 good
**Presentation:** 3 good
**Contribution:** 3 good
**Rating:** 6
**Confidence:** 2

**Summary:**

This manuscript proposes a method of incorporating conservation laws into neural operators (clawNOs) that perform physical modeling. Using the proposed method, high accuracy can be achieved, especially for small sample sizes. The proposed method is demonstrated by applying it to material deformation, fluid dynamics, and atmospheric simulations.

**Strengths:**

This manuscript proposes a novel method of incorporating physical conservation laws into the Neural Operator architecture, enabling physical modeling with small data sizes. It also validates the proposed method on a wide range of physics system and demonstrates that it outperforms existing state-of-the-art Neural Operators. As such, I believe that the proposed method is generally useful in physical modeling.

**Weaknesses:**

Lack of comparison with existing methods other than neural operators. There is no examination of the scalability of the method in high dimensional systems. It is unclear how the computational cost changes as the conservation laws and system dimensionality increase in complexity.

**Questions:**

It would be good to compare the method with existing methods other than neural operators.
It would be good to add an examination of the scalability of the methods in high dimensional systems.
It would be good to present how the computational cost changes as the conservation laws and the dimensionality of the system become more complex.

---

> ### Author Response · Authors · 2023-11-20
> **Response, part 1**
>
> We thank the reviewer for their valuable time and insightful suggestions. We have added an non-neural-operator baseline per the reviewer's suggestion. Our response:
>
> **Non-neural-operator baseline**: We have added a state-of-the-art non-neural-operator baseline, Lagrangian neural networks (LNN [1-2]), in the incompressible Navier-Stokes (NS) experiment. While both LNN and clawNO hard encode conservation laws in their model architecture, the fundamental difference between neural operators (NO) and non-NO networks is that NOs can learn from a certain resolution of the solution and infer on different resolutions, which makes them particularly suitable for PDE solving problems. The results of LNN on the incompressible NS dataset is included in table 1 in the revised manuscript. We observe that LNN seemingly cannot learn the correct solution as the test error remains above 20% across all three data regimes. This is due to the fact that LNN is designed to take the current position and velocity information as input and the acceleration as output. Then, the predicted acceleration is used to update the velocity and position in the next time instance. Since this dataset is relatively sparse in time (the time step in the dataset is $4\times10^4$ times larger than what is employed in the numerical solver for data generation), it leads to large errors from temporal integration in LNNs. On the other hand, all neural operator architectures parameterize the neural network as a function-to-function mapping. In this context, they learn the mapping from previous velocity to the future velocity directly. Therefore, we can see that all neural operator architectures do not suffer from this temporal integration error.
>
> [1] Cranmer, Miles, et al. "Lagrangian neural networks." arXiv preprint arXiv:2003.04630 (2020).
>
> [2] Müller, Eike Hermann. "Exact conservation laws for neural network integrators of dynamical systems." Journal of Computational Physics 488 (2023): 112234.

---

> ### Author Response · Authors · 2023-11-20
> **Response, part 2**
>
> **Scalability in high dimensional systems and model complexity**: The complexity of our clawNOs is very simular to their NO counterparts. Taking clawFNOs as an example, the trainable part of the clawNO model consists of two fixed-size MLPs for the lifting layer and the projection layer, and $L$ numbers of Fourier layers between them. Denote $d_u$ as the input function dimension, $H$ as the latent dimension after lifting, $M$ as the total number of grids, $m$ as the number of Fourier modes on each dimension after truncation (which is often taken as half of the number of grids in each dimension, $M^{1/p}$, in practice), and $p$ as the problem dimension. During the lifting layer a vector-valued input function taking values in $\mathbb{R}^{d_u}$ is linearly and locally mapped to the first-layer feature function $h(\cdot,0)$ taking values in $\mathbb{R}^{H}$, and hence the number of trainable parameters is $Hd_u+H$. Then, each iterative Fourier layer involves the integral with a trainable kernel weight matrix in the Fourier domain, which is of size $2H^2m^p=2^{1-p}H^2M$, and a local linear transformation which involves $H^2+H$ numbers of trainable parameters. Then, the last iterative layer feature function, $h(\cdot,L)$, is projected to the skew symmetric matrix-valued function $\mu$ with a two-layer MLP.  Since the skew-symmetric matrix-valued function $\mu$ is of degree of freedom $p(p-1)/2$ at each point x, the projection layer maps a size $H$ input vector to a size $p(p-1)/2$ vector. Assume that the hidden layer of this MLP is of $d_Q$ neurons, the total number of trainable parameters in projection layer will be $Hd_Q+d_Q+d_Qp(p-1)/2+p(p-1)/2$. Finally, $\mu$ will go through the pre-calculated differentiation layer, with $p^2(p-1)M^2/2=p^2(p-1)m^{2p}/2$ numbers of non-trainable parameters in the FNO case. From the above calculation, we can see that clawFNO involves $(d_Q+H(d_u+d_Q+L+1)+LH^2)+\dfrac{(d_Q+1)}{2}p(p-1)+2LH^2m^p$ numbers of trainable parameters, while the vanilla FNO involves $(d_Q+H(d_u+d_Q+L+1)+LH^2)+(d_Q+1)p+2LH^2m^p$ numbers of trainable parameters. Therefore, the number of trainable parameters in clawFNO and FNO only differs in the second part of their projection layer, where clawFNO has $\dfrac{(d_Q+1)}{2}p(p-1)$ numbers of parameters and FNO has $(d_Q+1)p$. When $p>3$, clawFNO will have a larger number of trainable parameters. However, we want to point out that since the number of parameter in the iterative layer ($2LH^2m^p$) grows exponentially with dimension $p$, it dominates the computational cost, and the differences between clawFNO and FNO are negligible. This fact is numerically verified in table 5 of the revised manuscript, where one can see that the number of trainable parameters and the GPU cost of clawNOs and their counterparts are almost the same. During the inference, the non-trainable parameters in clawNO will play a role and we therefore observe a minor increase in inference time. We have added the above discussion in Appendix B.1 of the revised manuscript, as a remark following the analysis for table 5.
>
> Thank you once again, for your favorable assessment of our work and your excellent suggestions. We would be very happy to answer follow-up questions or follow any additional suggestions.

---

### Official Review · Reviewer_UzYu · 2023-11-07

**Soundness:** 1 poor
**Presentation:** 1 poor
**Contribution:** 2 fair
**Rating:** 3
**Confidence:** 5

**Summary:**

This paper introduces conservation law-encoded neural operators (clawNOs) that allows satisfaction of conservation laws by providing a divergent free prediction. The proposed method can be integrated with any neural operator architecture to enforce the conservation law on model prediction. To evaluate the performance of clawNO models different PDE-based problems were considered and the model accuracy was compared with a few baseline models such as vanila FNO, GFNO. The numerical experiments demonstrated that the clawNOs can outperform the baseline methods specially when there is limited number of training sample data is available.

**Strengths:**

- The idea of baking conservation law into NO model architecture is a valuable effort towards more reliable ML models that respect physics-based constraints such as conservation laws.
- the variety of scientific problems considered for the evaluation of the proposed method is nice. Also providing the effect of different size of training data was insightful and nice.

**Weaknesses:**

The following are the main reasons that I do not recommend the publication of this paper:
- The presentation of paper could have been much better. For some parts such as last two paragraphs of section 3.1 and section 3.2 sound very unclear to me. Additionally, the description of different experiments and the result discussions could have been better. for example: were all tests out of domain evaluation or in-domain evaluation? for the cases where FNO outperformed GFNO, authors could easily provide numerical investigation instead of describing the reason as their "guess" (I refer to last paragraph of section 4.1)
- Another observation is that there are missing related papers which aimed to address the same topic (enforcing conservation laws), which could potentially raise questions about the comprehensiveness of this work, the literature review, and the originality of the idea. For example, one closely relates work is:
Neural Conservation Laws: A Divergence-Free Perspective (https://arxiv.org/abs/2210.01741)
Since the idea of this paper is pretty much similar to this past work, authors should have at least acknowledged that the idea of thsi paper is related. It would be more appropriate to have discussions about the similarities and the parts the methods departs, and why not using this work as one of the baselines? There are of course additional works that could have been at least covered in literature review, such as: learning physical models that can respect conservation laws (https://arxiv.org/abs/2302.11002) which proposes a projection method to enforce integral form of conservation, learning differential solvers for systems with hard constraints (https://arxiv.org/pdf/2207.08675.pdf), and definitely more!
- I wish authors also had provided information (preferably numerical experiments) to discuss the computational complexity of their method. As far as I found, they just mentioned in one sentence that their method has comparable network size and computational cost. A good quality works requires to explore different aspects and report them properly.

**Questions:**

- Is this method applicable for other NO models such as Koopman neural operators (KNO), which has been gaining attention for learning long term dynamics?
- Does enforcing the divergence-free into model architecture bring any challenges for model training? could it cause numerical stability? could you comment on this?
- I know that the commonly used NO architecture is FNO, even for non-periodic boundary conditions. But how about using other basis functions which are shown to be more appropriate choices for other boundary constraints in conventional spectral methods, such as Chebychev for Dirichlet? It would be nice to address this as a possibility, and potential challenges which made authors not try that?
-  As mentioned in weakness section, could you comment on the computational complexity and potential challenges you had in your experiments?
- (very minor comment) For Section 4.1, keep in mind that vorticity is a vector variable, just as velocity field, and for the 2D problem considered it only has the component normal to page.
- I did not clearly understand how training data and evaluation data are related? in other words, did authors conduct out of domain evaluation (which is more challenging task even for NO models), or it was in-domain evaluation. Could you clarify your train-test split and provide some details on the evaluation procedure?
- It is great that authors ran replicates of the same problem (with different seed of course) and reported mean and standard deviation. But how did they decide that 3 replicates could be enough? Did you try more replicates for at least one problem to see how the deviation change?
- In last paragraph of section 4.1, for GFNO lower performance you brought a "possible" reason that: This is possibly due to the expressivity degradation in GFNO outweighing the equivariant advantage, as the latent width of GFNO is reduced from 20 to 11 in order to maintain a similar total number of parameters to FNO for fair comparison.
Did you try the larger latent width to confirm that? It would be nice that you provide numerical reasoning.

**Details Of Ethics Concerns:**

No concern.

---

> ### Author Response · Authors · 2023-11-20
> **Response, part 1**
>
> We thank the reviewer for their valuable time and constructive comments. We have added new baselines, new tests, and new clarifications based on the reviewer's suggestions. Our response:
>
> **Clarifications of sections 3.1 and 3.2**: We thank the reviewer's valuable suggestion and have added more clarifications in the appendix (highlighted in red), due to the page limit. In particular, to clarify section 3.1 we have added a section in appendix A for the necessary definition and detailed derivations in differential forms. We have also provided the detailed derivations of section 3.2 in appendix D.
>
> **Related papers and our novelty**: We thank the reviewer's suggestion. We would like to point out that we have cited [1] and discussed the differences and limitations in their setting, in the last paragraph of Section 2 in the original manuscript. Per the reviewer's suggestion, we have also added [2] and [3] and discussed the differences between their approaches and our approach. Generally speaking, existing architectures [1-3] can only be applied on the known PDE-solving scenario where the full governing law is given, while in our framework we consider the physics discovery scenario where the constitutive laws are hidden and the model has to be learned from data. As a result, our approach can handle experimental data and is generalizable to different input functions, e.g., different initial conditions and boundary conditions. Another major difference (which is also the novelty in our architecture) is: in [1] the PINN architecture is used to approximate the solutions of this particular governing equation, i.e., mapping from x to the solution u(x). As such, the differential forms are approximated with automatic differentiation in NNs trivially. In our neural operator architecture, the differential forms can no longer be approximated via automatic differentiation, and therefore we propose a new architecture by attaching a pre-calculated numerical differentiation layer after the projection layer and provide error estimates in Section 3.2. We have added the above discussion, together with more details of [1-3], to the last paragraph of Section 2.
>
> We would also like to point out that the novelty of our idea is NOT on formulating the conservation law into a divergence-free architecture with a differential form representation. In fact, such a representation is well-known for more than a decade, see, e.g., [4], way before [1]. We have added more references on differential forms in the revised manuscript, to avoid such a confusion. The novelty of our approach is 1) the combination of neural operator with the mass conservation as a basic physical law, to learn the unknown physical system responses such as the constitutive laws with improved data efficiency; and 2) the design of a pre-calculated numerical differentiation layer in the neural operator architecture, to enable the incorporation of differential form-guaranteed divergence-free output, without increasing the number of trainable parameters and computational cost. To our best knowledge, neither of these two ideas have been discussed in existing neural operator literature.
>
> [1] Richter-Powell, Jack, Yaron Lipman, and Ricky TQ Chen. "Neural conservation laws: A divergence-free perspective." Advances in Neural Information Processing Systems 35 (2022): 38075-38088.
>
> [2] Négiar, Geoffrey, Michael W. Mahoney, and Aditi Krishnapriyan. "Learning differentiable solvers for systems with hard constraints." The Eleventh International Conference on Learning Representations. 2022.
>
> [3] Hansen, Derek, et al. "Learning Physical Models that Can Respect Conservation Laws." ICLR 2023 Workshop on Physics for Machine Learning. 2023.
>
> [4] Barbarosie, Cristian. "Representation of divergence-free vector fields." Quarterly of applied mathematics 69.2 (2011): 309-316.
>
> **Reason for FNO outperforming GFNO**: We appreciate the reviewer's insightful suggestion, and have performed an experiment where GFNO's latent dimension is increased to match the latent dimension of FNO. The resulting error is decreased to 1.19\% in the large data regime, which is on a similar scale to FNO (1.47\%). In this setting, the number of trainable parameters in GFNO is increased to 3,412,142. This thus verifies our guess. We have included this discussion in the last paragraph of section 4.1 in the revised manuscript.
>
> **Koopman neural operator**: Technically, our proposed method is realized by adding an additional differentiation layer in the neural operator, and hence it is applicable to all neural operator architectures. In fact, it will be an interesting future direction to combine our method with KNOs, which would possibly further improve the generalizability in small data and long-term integration regimes. In the revised manuscript, we have included KNO as an additional state-of-the-art NO baseline in our incompressible Navier-Stokes experiment.

---

> ### Author Response · Authors · 2023-11-20
> **Response, part 2**
>
> **Out-of-domain/in-domain evaluation**: Our small data regime setting is designed to test the performance of the trained models when the available data is scarce and does not provide a good coverage of the parameter range (i.e., out-of-domain). Taking the incompressible Navier-Stokes experiment as an example, when ntrain is as small as $10$, it is expected that some test samples will be out of the training regime. To validate this, in the paper revision we show in Fig. 3 of the appendix the data distribution in the incompressible Navier-Stokes experiment case 1. In particular, the histogram of the solution L2-norms are plotted in the $x$ and $y$ components. One can see that in the small data regime, the training set does not provide a good coverage of the dataset, and a large number of test samples are outside the training regime. Then, to illustrate the generalizability of clawNOs, we also plotted the per-sample test errors in Fig. 3a, where one can see that clawNO's error is much smaller than that of the non-claw counterpart, especially in the out-of-distribution regime. The training dataset of the medium and large datasets provide good coverage of the entire dataset and are aimed to test the in-domain predictability of various models.
>
> **Other basis functions**: Our formulation is independent of the basis functions used in the iterative layers. Our pre-calculated differentiation layer can be added after any neural operator to impose the divergence-free constraint. Therefore, our approach is readily applicable to a Chebychev neural operator. Here, FNO is considered because FFT is generally much faster. We have also demonstrated the applicability of our formulation in the GNO based setting in addition to FNO.
>
> **Challenge for training and numerical stability**: We have not observed any particular challenge in training or any numerical stability issue from clawNOs. The training and testing procedures are the same as all other NO baselines. This is because our divergence-free constraint is hard-coded into the model architecture through a pre-calculated layer, which makes it fundamentally different from the soft constraint approaches where a penalization term is added into the training loss (see, e.g., PINO [5]). As reported in [6], the soft constraint approach may result in a deteriorated training loss landscape, and face challenges in training.
>
> [5] Li, Zongyi, et al. "Physics-informed neural operator for learning partial differential equations." arXiv preprint arXiv:2111.03794 (2021).
>
> [6] Gopakumar, Vignesh, Stanislas Pamela, and Debasmita Samaddar. "Loss landscape engineering via data regulation on PINNs." Machine Learning with Applications 12 (2023): 100464.
>
> **Computational complexity and computational cost**: A standard FNO model consists of two fixed-size MLPs for the lifting layer and the projection layer, and $L$ numbers of Fourier layers between them. The kernel weight matrix in the Fourier domain takes up the majority of the model size, requiring $O(LH^2m^p)$ trainable parameters, where $L$ is the number of Fourier layers, $H$ is the latent dimension after lifting, $m$ is the number of Fourier modes on each dimension after truncation, and $p$ is the problem dimension. Since the differentiation layer is pre-computed and does not require any training, the complexity of clawNOs is the same as that of a standard FNO. To further verify this analysis and compare the computational cost, we have provided detailed performance comparison of all models in table 5 of the revised manuscript. The comparison metrics include the total number of model parameters, the per-epoch runtime, inference time, the peak GPU memory usage, and the L2-norm of the output divergence. One can see that the number of trainable parameters of clawNOs as well as GPU memory usage remain the same as their counterparts. Similar observation also applies to the runtime in large data regime. The inference time has a minor increase, due to the fact that the inferred solution will go through the an additional layer which avoidably increases the computational cost.

---

> ### Author Response · Authors · 2023-11-20
> **Response, part 3**
>
> **More replicates**: We have followed the practices in existing NO literature (e.g., GFNOs) and performed 3 replicates of each experiment. To respond to the reviewer's question, we have performed 5 replicates of case 1 of the incompressible Navier-Stokes dataset, and the results are listed in the following table. One can see that the results are very similar to the ones using 3 seeds.
>
>
> *Test errors for the incompressible Navier--Stokes problem with the the relative mean squared error as a percentage averaged over five random seeds in each case. Here bold numbers highlight the best method.*
> | Case               || 2D models | #param || ntrain=10 || ntrain=100 || ntrain=1000 |
> | :------------------ | ---| :--------: | :---------: |---| :--------: |---|:--------: |---| :-----------: |
> | $T_{in}$ = 10, $T$ = 20 || clawGFNO | 853,231 || 12.64%$\pm$1.10% || **4.80%$\pm$0.32%** || **1.22%$\pm$0.07%** |
> | $T_{in}$ = 10, $T$ = 20 || clawFNO  | 928,861 ||**12.43%$\pm$0.76%**|| 5.33%$\pm$0.40%   || 1.46%$\pm$0.08%     |
> | $T_{in}$ = 10, $T$ = 20 || GFNO-p4  | 853,272 || 28.82%$\pm$3.42%   || 7.82%$\pm$0.41%   || 3.06%$\pm$0.66%     |
> | $T_{in}$ = 10, $T$ = 20 || FNO      | 928,942 || 22.68%$\pm$1.42%   || 5.76%$\pm$0.34%   || 1.54%$\pm$0.17%     |
> | $T_{in}$ = 10, $T$ = 20 || UNet     | 920,845 || 53.12%$\pm$9.12%   || 16.89%$\pm$1.81%  || 5.98%$\pm$0.40%     |
> | $T_{in}$ = 10, $T$ = 20 || LSM      |1,211,234|| 31.43%$\pm$1.58%   || 13.51%$\pm$0.90%  || 5.53%$\pm$0.19%     |
> | $T_{in}$ = 10, $T$ = 20 || UNO      |1,074,522|| 21.94%$\pm$1.84%   || 11.08%$\pm$0.54%  || 5.36%$\pm$0.12%     |
> | $T_{in}$ = 10, $T$ = 20 || KNO      | 890,055 || 19.40%$\pm$2.12%   || 11.60%$\pm$2.22%  || 9.63%$\pm$1.44%     |
>
>
>
> Thank you very much, once again, for your excellent suggestions. We would be very happy to answer any follow-up or additional questions you may have.

---

### Official Review · Reviewer_1SzR · 2023-11-08

**Soundness:** 3 good
**Presentation:** 2 fair
**Contribution:** 3 good
**Rating:** 6
**Confidence:** 2

**Summary:**

The paper targets a hot topic, where a group of deep learning that encodes inductive bias to recover physical governing functions. Specifically, the conservation law is integrated into neural networks for physical consistency. Different from existing work, this work builds the conservation law into the networks and considers a divergence-free aspect.

**Strengths:**

Considering physics-based learning is a hot topic with plenty of work, the paper provides a detailed review and a smooth transition from existing work to the proposed work.

The built-in conservation law is novel, enabling a hard constraint in data-driven function discovery.

**Weaknesses:**

Except for the neural operators adopted as baselines, there are other deep learning methods using physics-based inductive biases, such as energy conservation in Hamiltonian/Lagrangian neural networks and their variations. In methodology, it’s unclear what is the distinction of the proposed method. In experiments, there is no comparison with these methods to show the strengths.

**Questions:**

Are the proposed methods free of any prior knowledge of the targeted physical functions? Could the author(s) explain why, compared to previous designs?

The conservation law encoded by this work seems generalizable for different conservative quantities in various physical systems. Is it true?

---

> ### Author Response · Authors · 2023-11-20
>
> We thank the reviewer for their time and valuable suggestions. We have added the Lagrangian neural network as an additional baseline and provide additional clarification regarding the prior knowledge, based on the reviewer's suggestion. Our response:
>
> **Non-neural-operator baseline**: We have added a state-of-the-art non-neural-operator baseline, Lagrangian neural networks (LNN) [1,2], in the incompressible Navier-Stokes (NS) experiment. The results are provided in table 1 in the revised manuscript. We observe that LNN seemingly cannot learn the correct solution as the test error remains above 20\% across all three data regimes. This is due to the fact that LNN is designed to take the current position and velocity information as input functions and the acceleration as output function. Then, the predicted acceleration is used to update the velocity and position in the next time instance. Since this dataset is relatively sparse in time (the time step in the dataset is $4\times10^4$ times larger than what is employed in the numerical solver for data generation), it leads to large errors from temporal integration in LNNs. On the other hand, all neural operator architectures parameterize the neural network as a function-to-function mapping. In this context, they learn the mapping from previous velocity to the future velocity directly. Therefore, we can see that all neural operator architectures do not suffer from this temporal integration error.
>
> [1] Cranmer, Miles, et al. "Lagrangian neural networks." arXiv preprint arXiv:2003.04630 (2020).
>
> [2] Müller, Eike Hermann. "Exact conservation laws for neural network integrators of dynamical systems." Journal of Computational Physics 488 (2023): 112234.
>
> **Prior knowledge of targeted physical functions**: In clawNOs, the only knowledge is on which quantity should be conserved, i.e., equation (5) in the main text. They are completely free of any other prior knowledge about the targeted physical system, and hence are readily applicable to discover unknown physical models from data. Taking the fluid dynamics problem as an example, we know that the mass should be conserved, but the constitutive law, i.e., the law governing fluid velocity, pressure, and the detailed formulation of viscous stress tensor-strain rate relationship, remains unknown. As a result, our setting can be applied to generic fluid problems from experimental measurements, no matter if it is a Newtonian fluid or a viscoelastic flow. This property is realized by employing the neural operator architecture [3], which learns a function-to-function mapping directly from data, and does not require prior knowledge. That means, the additional physical laws are learned from data.
>
> On the other hand, in the previous design of divergence-free neural networks, e.g., [4], it was assumed that the full physics is known. This is because [4] is formulated based on the physics-informed neural network (PINN, see [5]) setting, where the training loss is formulated based on the governing law without data. Therefore, the full governing system is required to guarantee the well-posedness of this PDE-solving problem.
>
> [3] Kovachki, Nikola, et al. "Neural operator: Learning maps between function spaces." arXiv preprint arXiv:2108.08481 (2021).
>
> [4] Richter-Powell, Jack, Yaron Lipman, and Ricky TQ Chen. "Neural conservation laws: A divergence-free perspective." Advances in Neural Information Processing Systems 35 (2022): 38075-38088.
>
> [5] Raissi, Maziar, Paris Perdikaris, and George E. Karniadakis. "Physics-informed neural networks: A deep learning framework for solving forward and inverse problems involving nonlinear partial differential equations." Journal of Computational physics 378 (2019): 686-707.
>
>
> **Encoded conservation law being generalizable for different conservative quantities**: Yes, our encoded conservation law is generalizable for different conservative quantities as long as they satisfy the continuity equation.
>
> Thank you once again, for your favorable assessment of our work and your excellent suggestions. We would be very happy to answer follow-up questions or follow any additional suggestions.

---

### Official Review · Reviewer_pjQT · 2023-11-13

**Soundness:** 2 fair
**Presentation:** 2 fair
**Contribution:** 3 good
**Rating:** 5
**Confidence:** 2

**Summary:**

The paper proposes a way to automatically guarantee the divergence-free of the NO outputs. This is done by introducing additional layers (the skew-symmetric matrix  and numerical differentiation layer) after the regular NO layers to form. The divergence-free architecture can be applied to different kinds of NO architectures.
The proposed method is evaluated on several examples (elasticity, shallow water equations and incompressible Navier-Stokes equations.) and compared with several existing methods (INO, GNO, FNO, G-FNO, U-Net). From the results, the method performs better compared with others, especially on small training dataset.

**Strengths:**

- Enforcing the conservation law can be a very useful way to improve the performance of neural operators, and the paper proposes a novel way to construct a network architecture to enforce a divergence-free output of NOs.
(I am not an expert in this area and I am assuming the paper’s method is the first one to enforcing the divergence-free form using the skew-symmetric matrix and numerical differentiation layers, based on their discussion of related work.)

**Weaknesses:**

- The examples provided in the paper are relatively simple. There are not enough evaluation on real 3D models.  It would be good to see how the method performs on more complex cases, for example, 3D NS equations.
- The paper doesn’t explain the setup of experiments well. For example, different test cases use different Ts, number of samples and other settings. Why different settings were chosen for each test case?

**Questions:**

- As the paper proposes an architecture to enforce the divergence-free NO output, it would be good to have some analysis of the output divergence of the proposed method compared to other methods, and whether the improvement of the output divergence corresponds to the improvement of the accuracy of the proposed method.
- It would be good if the paper could provide more details in the training experiments. For example, how many training iterations it takes for the proposed methods and other methods. How much time it takes for each iteration in training and inference.

---

> ### Author Response · Authors · 2023-11-20
>
> We thank the reviewer for their time and valuable suggestions. Our response:
>
> **Applicability to real 3D models**: We have followed other successful neural operator papers (FNO, G-FNO) and two popular PDE benchmark papers (PDEBench, PDEArena) in terms of experimental design. Since these experiments are well established and widely accepted, it is more transparent to perform comparison against baselines and demonstrate the performance of clawNO. While we acknowledge that we did not include (spatially) 3D models due to the availability of datasets and the restriction of time during the rebuttal period, we will try to investigate 3D examples in the future. Moreover, we want to point out that clawNOs are immediately applicable to more complex examples. For instance, the atmospheric modeling of the entire globe manifests significant complexity.
>
> **Analysis of output divergence**: We have computed the averaged L2-norm of the output divergence (the sum of the L2-norm of the divergence of all the test samples divided by the number of test samples) of both clawNOs and the selected baselines in the incompressible Navier-Stokes experiment, and included the results in table 5 of the revised manuscript. As shown in the table, both clawGFNO and clawFNO lead to very small divergence of the output, whereas other non-claw models all have significantly larger divergence in the output. Given that the only architectural difference between clawGFNO and GFNO (and between clawFNO and FNO) is the differentiation layer that enforces a divergence-free prediction and that clawNOs significantly outperform the non-claw counterparts in prediction accuracy, the divergence-free architecture of clawNOs correlates strongly with the improvement of the model accuracy.
>
> **More details on experiments**: The experimental details can be found in the appendix (A1-A4 corresponding to the four experiments) given the page limitation of the main text. We also include more details on the training strategies in Appendix A5. Specifically, all the 2D models (only two spatial dimensions) are trained for a total of 100 epochs whereas all the 3D models (two spatial dimensions plus one temporal dimension) are trained for 500 epochs with an early stop if the validation loss stops improving for consecutive 100 epochs. 2D models are trained with less number of epochs as one training sample in 3D corresponds to $(T-T_{in})$ training samples in 2D. A detailed performance breakdown for all the models, including the total number of model parameters, the per-epoch runtime during training, the inference time, the peak GPU memory usage, and the L2-norm of the output divergence, is provided in table 5 of the appendix in the revised manuscript. In terms of the different settings for each test case, we have followed the settings of previously published and successful neural operator baselines and two PDE benchmark papers (PDEBench and PDEArena) and adopted the same settings as theirs with only two exceptions: The first exception is in the incompressible NS experiment where we decide to add one more challenging setting of $(T_{in}=5, T_{out}=25)$ in addition to the commonly adopted $(T_{in}=10, T_{out}=20)$ to further analyze the model accuracy under a longer time extrapolation. The 2nd exception is in the atmospheric modeling experiment, for which we have explained our choice of the setting in Appendix A3.
>
> Thank you once again, for your excellent points. Please let us know if you have any follow-up questions or further suggestions. We would be happy to continue the discussion.

---

### Author Response · Authors · 2023-11-19
**Thank you!**

We thank the reviewers for the constructive comments, for recognizing the importance/usefulness of our work (reviewers pjQT, UzYu, t5ea, L28W), the novelty of clawNO's architecture (reviewers 1SzR, pjQT, t5ea), and the large variety of experimental evaluation (reviewers UzYu, t5ea).

Per the reviewers' suggestions, we have added three additional neural operator baselines (i.e., Koopman Neural Operators [1], Latent Spectral Models [2], and U-shaped Neural Operators [3]) and a non-neural-operator baseline (Lagrangian Neural Networks [4]) in the incompressible Navier-Stokes experiment. Our original conclusion still stands: clawNOs consistently outperform all selected baselines in all three considered data regimes. We have also added new references, and discussed the differences between the proposed method with existed divergence-free neural network architectures [5,6]: existed architectures only focus on the PDE solving scenario where the full governing law is given, while in our framework we consider the physics discovery scenario where the constitutive laws are hidden and the model has to be learnt from data. Another major difference (which is also the novelty in our architecture) is: in [5] the PINN architecture is used to approximate the solutions of each particular governing equation, i.e., mapping from x to the solution u(x). As such, the differential forms are approximated with automatic differentiation in NNs. In our neural operator architecture the differential forms can no longer be approximated via automatic differentiation, and therefore we propose a new architecture by attaching a pre-calculated numerical differentiation layer after the projection layer.

[1] Xiong, Wei, et al. "Koopman neural operator as a mesh-free solver of non-linear partial differential equations." arXiv preprint arXiv:2301.10022 (2023).

[2] Wu, Haixu, et al. "Solving High-Dimensional PDEs with Latent Spectral Models." arXiv preprint arXiv:2301.12664 (2023).

[3] Rahman, Md Ashiqur, Zachary E. Ross, and Kamyar Azizzadenesheli. "U-no: U-shaped neural operators." arXiv preprint arXiv:2204.11127 (2022).

[4] Müller, Eike Hermann. "Exact conservation laws for neural network integrators of dynamical systems." Journal of Computational Physics 488 (2023): 112234.

[5] Richter-Powell, Jack, Yaron Lipman, and Ricky TQ Chen. "Neural conservation laws: A divergence-free perspective." Advances in Neural Information Processing Systems 35 (2022): 38075-38088.

[6] Hansen, Derek, et al. "Learning Physical Models that Can Respect Conservation Laws." ICLR 2023 Workshop on Physics for Machine Learning. 2023.

We invite the reviewers to check the additional results in the revised manuscript, where the revised parts are highlighted *in red*.

---

### Author Response · Authors · 2023-11-22
**Thank you again.**

Dear AC and reviewers,

We are really grateful and cordially thank all of you for the effort and time you spent on our paper. Your comments undoubtedly helped to improve our work. We also tried our best to address the raised concerns.

Could you please let us know if there is anything that we can do to further improve the paper? We are prepared to answer any of your questions.

Thank you.

-The authors

---

### Meta-Review · Area_Chair_ZgRa · 2023-12-07

**Metareview:**

The paper explores the application of constraining neural operators (NO) to adhere to fundamental conservation laws, such as mass or volume conservation. These conservation laws, leading to a continuity equation, can be expressed as a divergence-free vector field. The authors introduce the use of differential forms to construct divergence-free neural operators and present an implementation integrated into existing NOs. This is achieved through a numerical differential layer that enforces the divergence-free condition on the output of a NO. Experimental evaluations are conducted on four families of partial differential equations (PDEs), demonstrating improved performance compared to baseline NOs.

The reviewers raised concerns about the originality of the proposed method, noting that similar ideas of imposing physical constraints for neural solvers have been presented using differential forms. In response, the authors clarified that previous contributions assumed knowledge of the underlying equation describing the physical system, whereas their approach focuses on plain data-based methods without leveraging known PDEs. The novelty lies in proposing a solution for learning divergence-free operators, ensured by a numerical differentiation layer. Additionally, the authors addressed reviewers' requests by incorporating new baseline comparisons.

Even though the authors considerably clarified the first version of the paper, the reviewers did not raise their score to acceptance. We recommend a resubmission that directly incorporates all the clarifications and additions provided during the rebuttal

**Justification For Why Not Higher Score:**

The reviewers still have concerns

**Justification For Why Not Lower Score:**

a

---

### Decision · Program_Chairs · 2024-01-16

Reject